# THE NEURO-SYMBOLIC CONCEPT LEARNER: INTERPRETING SCENES, WORDS, AND SENTENCES FROM NATURAL SUPERVISION

**Jiayuan Mao**
MIT CSAIL and IIIS, Tsinghua University
mjy14@mails.tsinghua.edu.cn

**Chuang Gan**
MIT-IBM Watson AI Lab
ganchuang@csail.mit.edu

**Pushmeet Kohli**
Deepmind
pushmeet@google.com

**Joshua B. Tenenbaum**
MIT BCS, CBMM, CSAIL
jbt@mit.edu

**Jiajun Wu**
MIT CSAIL
jiajunwu@mit.edu

## ABSTRACT

We propose the Neuro-Symbolic Concept Learner (NS-CL), a model that learns visual concepts, words, and semantic parsing of sentences without explicit supervision on any of them; instead, our model learns by simply looking at images and reading paired questions and answers. Our model builds an object-based scene representation and translates sentences into executable, symbolic programs. To bridge the learning of two modules, we use a neuro-symbolic reasoning module that executes these programs on the latent scene representation. Analogical to human concept learning, the perception module learns visual concepts based on the language description of the object being referred to. Meanwhile, the learned visual concepts facilitate learning new words and parsing new sentences. We use curriculum learning to guide the searching over the large compositional space of images and language. Extensive experiments demonstrate the accuracy and efficiency of our model on learning visual concepts, word representations, and semantic parsing of sentences. Further, our method allows easy generalization to new object attributes, compositions, language concepts, scenes and questions, and even new program domains. It also empowers applications including visual question answering and bidirectional image-text retrieval.

## 1 INTRODUCTION

Humans are capable of learning visual concepts by jointly understanding vision and language (Fazly et al., 2010; Chrupała et al., 2015; Gauthier et al., 2018). Consider the example shown in Figure 1-I. Imagine that someone with no prior knowledge of colors is presented with the images of the red and green cubes, paired with the questions and answers. They can easily identify the difference in objects' visual appearance (in this case, color), and align it to the corresponding words in the questions and answers (Red and Green). Other object attributes (e.g., shape) can be learned in a similar fashion. Starting from there, humans are able to inductively learn the correspondence between visual concepts and word semantics (e.g., spatial relations and referential expressions, Figure 1-II), and unravel compositional logic from complex questions assisted by the learned visual concepts (Figure 1-III, also see Abend et al. (2017)).

Motivated by this, we propose the neuro-symbolic concept learner (NS-CL), which jointly learns visual perception, words, and semantic language parsing from images and question-answer pairs. NS-CL has three modules: a neural-based perception module that extracts object-level representations from the scene, a visually-grounded semantic parser for translating questions into executable programs, and a symbolic program executor that reads out the perceptual representation of objects, classifies their attributes/relations, and executes the program to obtain an answer.

---

Project page: http://nscl.csail.mit.edu

I. Learning basic, object-based concepts.

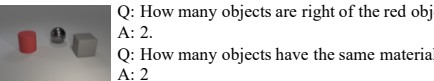

Q: What's the color of the object?
A: Red.
Q: Is there any cube?
A: Yes.

Q: What's the color of the object?
A: Green.
Q: Is there any cube?
A: Yes.

II. Learning relational concepts based on referential expressions.

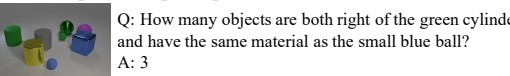

Q: How many objects are right of the red object?
A: 2.
Q: How many objects have the same material as the cube?
A: 2

III. Interpret complex questions from visual cues.

Q: How many objects are both right of the green cylinder and have the same material as the small blue ball?
A: 3

Figure 1: Humans learn visual concepts, words, and semantic parsing jointly and incrementally. **I.** Learning visual concepts (red *vs*. green) starts from looking at simple scenes, reading simple questions, and reasoning over contrastive examples (Fazly et al., 2010). **II.** Afterwards, we can interpret referential expressions based on the learned object-based concepts, and learn relational concepts (e.g., on the right of, the same material as). **III** Finally, we can interpret complex questions from visual cues by exploiting the compositional structure.

NS-CL learns from natural supervision (i.e., images and QA pairs), requiring no annotations on images or semantic programs for sentences. Instead, analogical to human concept learning, it learns via curriculum learning. NS-CL starts by learning representations/concepts of individual objects from short questions (e.g., What's the color of the cylinder?) on simple scenes (≤3 objects). By doing so, it learns object-based concepts such as colors and shapes. NS-CL then learns relational concepts by leveraging these object-based concepts to interpret object referrals (e.g., Is there a box right of a cylinder?). The model iteratively adapts to more complex scenes and highly compositional questions.

NS-CL's modularized design enables interpretable, robust, and accurate visual reasoning: it achieves state-of-the-art performance on the CLEVR dataset (Johnson et al., 2017a). More importantly, it naturally learns disentangled visual and language concepts, enabling combinatorial generalization w.r.t. both visual scenes and semantic programs. In particular, we demonstrate four forms of generalization. First, NS-CL generalizes to scenes with more objects and longer semantic programs than those in the training set. Second, it generalizes to new visual attribute compositions, as demonstrated on the CLEVR-CoGenT (Johnson et al., 2017a) dataset. Third, it enables fast adaptation to novel visual concepts, such as learning a new color. Finally, the learned visual concepts transfer to new tasks, such as image-caption retrieval, without any extra fine-tuning.

## 2 RELATED WORK

Our model is related to research on joint learning of vision and natural language. In particular, there are many papers that learn visual concepts from descriptive languages, such as image-captioning or visually-grounded question-answer pairs (Kiros et al., 2014; Shi et al., 2018; Mao et al., 2016; Vendrov et al., 2016; Ganju et al., 2017), dense language descriptions for scenes (Johnson et al., 2016), video-captioning (Donahue et al., 2015) and video-text alignment (Zhu et al., 2015).

Visual question answering (VQA) stands out as it requires understanding both visual content and language. The state-of-the-art approaches usually use neural attentions (Malinowski & Fritz, 2014; Chen et al., 2015; Yang et al., 2016; Xu & Saenko, 2016). Beyond question answering, Johnson et al. (2017a) proposed the CLEVR (VQA) dataset to diagnose reasoning models. CLEVR contains synthetic visual scenes and questions generated from latent programs. Table 1 compares our model with state-of-the-art visual reasoning models (Andreas et al., 2016; Suarez et al., 2018; Santoro et al., 2017) along four directions: visual features, semantics, inference, and the requirement of extra labels.

For visual representations, Johnson et al. (2017b) encoded visual scenes into a convolutional feature map for program operators. Mascharka et al. (2018); Hudson & Manning (2018) used attention as intermediate representations for transparent program execution. Recently, Yi et al. (2018) explored an interpretable, object-based visual representation for visual reasoning. It performs well, but requires fully-annotated scenes during training. Our model also adopts an object-based visual representation, but the representation is learned only based on natural supervision (questions and answers).

Anderson et al. (2018) also proposed to represent the image as a collection of convolutional object features and gained substantial improvements on VQA. Their model encodes questions with neural

| Models | Visual Features | Semantics | Extra Labels | | Inference |
| | | | # Prog. | Attr. | |
|---|---|---|---|---|---|
| FiLM (Perez et al., 2018) | Convolutional | Implicit | 0 | No | Feature Manipulation |
| IEP (Johnson et al., 2017b) | Convolutional | Explicit | 700K | No | Feature Manipulation |
| MAC (Hudson & Manning, 2018) | Attentional | Implicit | 0 | No | Feature Manipulation |
| Stack-NMN (Hu et al., 2018) | Attentional | Implicit | 0 | No | Attention Manipulation |
| TbD (Mascharka et al., 2018) | Attentional | Explicit | 700K | No | Attention Manipulation |
| NS-VQA (Yi et al., 2018) | Object-Based | Explicit | 0.2K | Yes | Symbolic Execution |
| NS-CL | Object-Based | Explicit | 0 | No | Symbolic Execution |

Table 1: Comparison with other frameworks on the CLEVR VQA dataset, w.r.t. visual features, implicit or explicit semantics and supervisions.

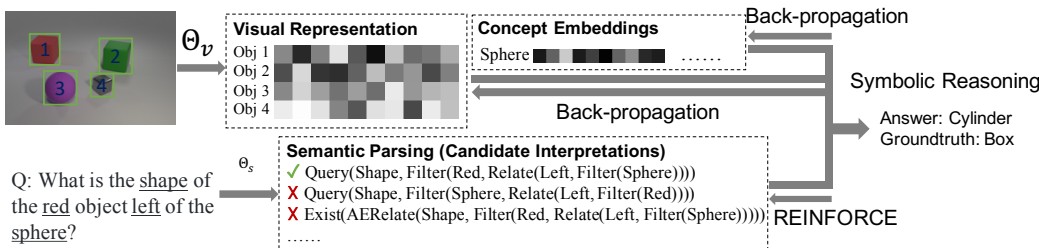

Figure 2: We propose to use neural symbolic reasoning as a bridge to jointly learn visual concepts, words, and semantic parsing of sentences.

networks and answers the questions by question-conditioned attention over the object features. In contrast, NS-CL parses question inputs into programs and executes them on object features to get the answer. This makes the reasoning process interpretable and supports combinatorial generalization over quantities (e.g., counting objects). Our model also learns general visual concepts and their association with symbolic representations of language. These learned concepts can then be explicitly interpreted and deployed in other vision-language applications such as image caption retrieval.

There are two types of approaches in semantic sentence parsing for visual reasoning: implicit programs as conditioned neural operations (e.g., conditioned convolution and dual attention) (Perez et al., 2018; Hudson & Manning, 2018) and explicit programs as sequences of symbolic tokens (Andreas et al., 2016; Johnson et al., 2017b; Mascharka et al., 2018). As a representative, Andreas et al. (2016) build modular and structured neural architectures based on programs for answering questions. Explicit programs gain better interpretability, but usually require extra supervision such as ground-truth program annotations for training. This restricts their application. We propose to use visual grounding as distant supervision to parse questions in natural languages into explicit programs, with *zero* program annotations. Given the semantic parsing of questions into programs, Yi et al. (2018) proposed a purely symbolic executor for the inference of the answer in the logic space. Compared with theirs, we propose a quasi-symbolic executor for VQA.

Our work is also related to learning interpretable and disentangled representations for visual scenes using neural networks. Kulkarni et al. (2015) proposed convolutional inverse graphics networks for learning and inferring pose of faces, while Yang et al. (2015) learned disentangled representation of pose of chairs from images. Wu et al. (2017) proposed the neural scene de-rendering framework as an inverse process of any rendering process. Siddharth et al. (2017); Higgins et al. (2018) learned disentangled representations using deep generative models. In contrast, we propose an alternative representation learning approach through joint reasoning with language.

## 3  NEURO-SYMBOLIC CONCEPT LEARNER

We present our neuro-symbolic concept learner, which uses a symbolic reasoning process to bridge the learning of visual concepts, words, and semantic parsing of sentences without explicit annotations

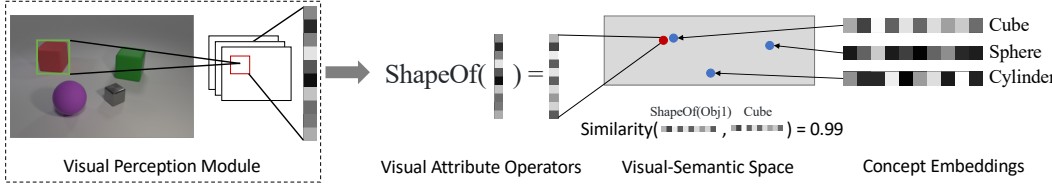

Figure 3: We treat attributes such as `Shape` and `Color` as neural operators. The operators map object representations into a visual-semantic space. We use similarity-based metric to classify objects.

for any of them. We first use a visual perception module to construct an object-based representation for a scene, and run a semantic parsing module to translate a question into an executable program. We then apply a quasi-symbolic program executor to infer the answer based on the scene representation. We use paired images, questions, and answers to jointly train the visual and language modules.

Shown in Figure 2, given an input image, the visual perception module detects objects in the scene and extracts a deep, latent representation for each of them. The semantic parsing module translates an input question in natural language into an executable program given a domain specific language (DSL). The generated programs have a hierarchical structure of symbolic, functional modules, each fulfilling a specific operation over the scene representation. The explicit program semantics enjoys compositionality, interpretability, and generalizability.

The program executor executes the program upon the derived scene representation and answers the question. Our program executor works in a symbolic and deterministic manner. This feature ensures a transparent execution trace of the program. Our program executor has a fully differentiable design w.r.t. the visual representations and the concept representations, which supports gradient-based optimization during training.

## 3.1 MODEL DETAILS

**Visual perception.** Shown in Figure 2, given the input image, we use a pretrained Mask R-CNN (He et al., 2017) to generate object proposals for all objects. The bounding box for each single object paired with the original image is then sent to a ResNet-34 (He et al., 2015) to extract the region-based (by RoI Align) and image-based features respectively. We concatenate them to represent each object. Here, the inclusion of the representation of the full scene adds the contextual information, which is essential for the inference of relative attributes such as size or spatial position.

**Concept quantization.** Visual reasoning requires determining an object's attributes (e.g., its color or shape). We assume each visual attribute (e.g., shape) contains a set of visual concept (e.g., `Cube`). In NS-CL, visual attributes are implemented as neural operators , mapping the object representation into an attribute-specific embedding space. Figure 3 shows an inference an object's shape. Visual concepts that belong to the shape attribute, including `Cube`, `Sphere` and `Cylinder`, are represented as vectors in the shape embedding space. These concept vectors are also learned along the process. We measure the cosine distances $\langle \cdot, \cdot \rangle$ between these vectors to determine the shape of the object. Specifically, we compute the probability that an object $o_i$ is a cube by $\sigma \left( \left( \langle \mathrm{ShapeOf}(o_i), v^{\mathtt{Cube}} \rangle - \gamma \right) / \tau \right)$, where $\mathrm{ShapeOf}(\cdot)$ denotes the neural operator, $v^{\mathtt{Cube}}$ the concept embedding of `Cube` and $\sigma$ the Sigmoid function. $\gamma$ and $\tau$ are scalar constants for scaling and shifting the values of similarities. We classify relational concepts (e.g., `Left`) between a pair of objects similarly, except that we concatenate the visual representations for both objects to form the representation of their relation.

**DSL and semantic parsing.** The semantic parsing module translates a natural language question into an executable program with a hierarchy of primitive operations, represented in a domain-specific language (DSL) designed for VQA. The DSL covers a set of fundamental operations for visual reasoning, such as filtering out objects with certain concepts or query the attribute of an object. The operations share the same input and output interface, and thus can be compositionally combined to form programs of any complexity. We include a complete specification of the DSL used by our framework in the Appendix A.

A. Curriculum concept learning

B. Illustrative execution of NS-CL

Figure 4: **A.** Demonstration of the curriculum learning of visual concepts, words, and semantic parsing of sentences by watching images and reading paired questions and answers. Scenes and questions of different complexities are illustrated to the learner in an incremental manner. **B.** Illustration of our neuro-symbolic inference model for VQA. The perception module begins with parsing visual scenes into object-based deep representations, while the semantic parser parse sentences into executable programs. A symbolic execution process bridges two modules.

Our semantic parser generates the hierarchies of latent programs in a sequence to tree manner (Dong & Lapata, 2016). We use a bidirectional GRU (Cho et al., 2014) to encode an input question, which outputs a fixed-length embedding of the question. A decoder based on GRU cells is applied to the embedding, and recovers the hierarchy of operations as the latent program. Some operations takes concepts their parameters, such as `Filter( Red )` and `Query( Shape )`. These concepts are chosen from all concepts appeared in the input question. Figure 4(B) shows an example, while more details can be found in Appendix B.

**Quasi-symbolic program execution.** Given the latent program recovered from the question in natural language, a symbolic program executor executes the program and derives the answer based on the object-based visual representation. Our program executor is a collection of deterministic functional modules designed to realize all logic operations specified in the DSL. Figure 4(B) shows an illustrative execution trace of a program.

To make the execution differentiable w.r.t. visual representations, we represent the intermediate results in a probabilistic manner: a set of objects is represented by a vector, as the attention mask over all objects in the scene. Each element, $\mathrm{Mask}_i \in [0, 1]$ denotes the probability that the $i$-th object of the scene belongs to the set. For example, shown in Figure 4(B), the first `Filter` operation outputs a mask of length 4 (there are in total 4 objects in the scene), with each element representing the probability that the corresponding object is selected out (i.e., the probability that each object is a green cube). The output "mask" on the objects will be fed into the next module (`Relate` in this case) as input and the execution of programs continues. The last module outputs the final answer to the question. We refer interested readers to Appendix C for the implementation of all operators.

### 3.2 TRAINING PARADIGM

**Optimization objective.** The optimization objective of NS-CL is composed of two parts: concept learning and language understanding. Our goal is to find the optimal parameters $\Theta_v$ of the visual

perception module Perception (including the ResNet-34 for extracting object features, attribute operators. and concept embeddings) and $\Theta_s$ of the semantic parsing module SemanticParse, to maximize the likelihood of answering the question $Q$ correctly:

$$\Theta_v, \Theta_s \leftarrow \arg\max_{\Theta_v, \Theta_s} \mathbb{E}_P[\Pr[A = \text{Executor}(\text{Perception}(S; \Theta_v), P)]], \qquad (1)$$

where $P$ denotes the program, $A$ the answer, $S$ the scene, and Executor the quasi-symbolic executor. The expectation is taken over $P \sim \text{SemanticParse}(Q; \Theta_s)$.

Recall the program executor is fully differentiable w.r.t. the visual representation. We compute the gradient w.r.t. $\Theta_v$ as $\nabla_{\Theta_v} \mathbb{E}_P[D_{\text{KL}}(\text{Executor}(\text{Perception}(S; \Theta_v), P)\|A)]$. We use RE-INFORCE (Williams, 1992) to optimize the semantic parser $\Theta_s$: $\nabla_{\Theta_s} = \mathbb{E}_P[r \cdot \log \Pr[P = \text{SemanticParse}(Q; \Theta_s)]]$, where the reward $r = 1$ if the answer is correct and 0 otherwise. We also use off-policy search to reduce the variance of REINFORCE, the detail of which can be found in Appendix D.

**Curriculum visual concept learning.** Motivated by human concept learning as in Figure 1, we employ a curriculum learning approach to help joint optimization. We heuristically split the training samples into four stages (Figure 4(A)): first, learning object-level visual concepts; second, learning relational questions; third, learning more complex questions with perception modules fixed; fourth, joint fine-tuning of all modules. We found that this is essential to the learning of our neuro-symbolic concept learner. We include more technical details in Appendix E.

## 4 EXPERIMENTS

We demonstrate the following advantages of our NS-CL. First, it learns visual concepts with remarkable accuracy; second, it allows data-efficient visual reasoning on the CLEVR dataset (Johnson et al., 2017a); third, it generalizes well to new attributes, visual composition, and language domains.

We train NS-CL on 5K images (<10% of CLEVR's 70K training images). We generate 20 questions for each image for the entire curriculum learning process. The Mask R-CNN module is pretrained on 4K generated CLEVR images with bounding box annotations, following Yi et al. (2018).

### 4.1 VISUAL CONCEPT LEARNING

**Classification-based concept evaluation.** Our model treats attributes as neural operators that map latent object representations into an attribute-specific embedding space (Figure 3). We evaluate the concept quantization of objects in the CLEVR validation split. Our model can achieve near perfect classification accuracy ($\sim$99%) for all object properties, suggesting it effectively learns generic concept representations. The result for spatial relations is relatively lower, because CLEVR does not have direct queries on the spatial relation between objects. Thus, spatial relation concepts can only be learned indirectly.

**Count-based concept evaluation.** The SOTA methods do not provide interpretable representation on individual objects (Johnson et al., 2017a; Hudson & Manning, 2018; Mascharka et al., 2018) . To evaluate the visual concepts learned by such models, we generate a synthetic question set. The diagnostic question set contains simple questions as the following form: "How many `red` objects are there?". We evaluate the performance on all concepts appeared in the CLEVR dataset.

Table 2 summarizes the results compared with strong baselines, including methods based on convolutional features (Johnson et al., 2017b) and those based on neural attentions (Mascharka et al., 2018; Hudson & Manning, 2018). Our approach outperforms IEP by a significant margin (8%) and attention-based baselines by >2%, suggesting object-based visual representations and symbolic reasoning helps to interpret visual concepts.

### 4.2 DATA-EFFICIENT AND INTERPRETABLE VISUAL REASONING

NS-CL jointly learns visual concepts, words and semantic parsing by watching images and reading paired questions and answers. It can be directly applied to VQA.

|  | Visual | Mean | Color | Mat. | Shape | Size |
|---|---|---|---|---|---|---|
| IEP | Conv. | 90.6 | 91.0 | 90.0 | 89.9 | 90.6 |
| MAC | Attn. | 95.9 | 98.0 | 91.4 | 94.4 | 94.2 |
| TbD (hres.) | Attn. | 96.5 | 96.6 | 92.2 | 95.4 | 92.6 |
| NS-CL | Obj. | **98.7** | **99.0** | **98.7** | **98.1** | **99.1** |

Table 2: We also evaluate the learned visual concepts using a diagnostic question set containing simple questions such as "How many `red` objects are there?". NS-CL outperforms both convolutional and attentional baselines. The suggested object-based visual representation and symbolic reasoning approach perceives better interpretation of visual concepts.

| Model | Visual | Accuracy (100% Data) | Accuracy (10% Data) |
|---|---|---|---|
| TbD | Attn. | 99.1 | 54.2 |
| TbD-Object | Obj. | 84.1 | 52.6 |
| TbD-Mask | Attn. | 99.0 | 55.0 |
| MAC | Attn. | 98.9 | 67.3 |
| MAC-Object | Obj. | 79.5 | 51.2 |
| MAC-Mask | Attn. | 98.7 | 68.4 |
| NS-CL | Obj. | **99.2** | **98.9** |

Table 3: We compare different variants of baselines for a systematic study on visual features and data efficiency. Using only 10% of the training images, our model is able to achieve a comparable results with the baselines trained on the full dataset. See the text for details.

Table 4 summarizes results on the CLEVR validation split. Our model achieves the state-of-the-art performance among all baselines using zero program annotations, including MAC (Hudson & Manning, 2018) and FiLM (Perez et al., 2018). Our model achieves comparable performance with the strong baseline TbD-Nets (Mascharka et al., 2018), whose semantic parser is trained using 700K programs in CLEVR (ours need 0). The recent NS-VQA model from Yi et al. (2018) achieves better performance on CLEVR; however, their system requires annotated visual attributes and program traces during training, while our NS-CL needs no extra labels.

Here, the visual perception module is pre-trained on ImageNet (Deng et al., 2009). Without pre-training, the concept learning accuracies drop by 0.2% on average and the QA accuracy drops by 0.5%. Meanwhile, NS-CL recovers the underlying programs of questions accurately ($> 99.9\%$ accuracy). NS-CL can also detect ambiguous or invalid programs and indicate exceptions. Please see Appendix F for more details. NS-CL can also be applied to other visual reasoning testbeds. Please refer to Appendix G.1 for our results on the Minecraft dataset (Yi et al., 2018).

For a systematic study on visual features and data efficiency, we implement two variants of the baseline models: TbD-Object and MAC-Object. Inspired by (Anderson et al., 2018), instead of the input image, TbD-Object and MAC-Object take a stack of object features as input. TbD-Mask and MAC-Mask integrate the masks of objects by using them to guide the attention over the images.

Table 3 summarizes the results. Our model outperforms all baselines on data efficiency. This comes from the full disentanglement of visual concept learning and symbolic reasoning: how to execute program instructions based on the learned concepts is programmed. TbD-Object and MAC-Object demonstrate inferior results in our experiments. We attribute this to the design of model architectures and have a detailed analysis in Appendix F.3. Although TbD-Mask and MAC-Mask do not perform better than the originals, we find that using masks to guide attentions speeds up the training.

Besides achieving a competitive performance on the visual reasoning testbeds, by leveraging both object-based representation and symbolic reasoning, out model learns fully interpretable visual concepts: see Appendix H for qualitative results on various datasets.

## 4.3 GENERALIZATION TO NEW ATTRIBUTES AND COMPOSITIONS

**Generalizing to new visual compositions.** The CLEVR-CoGenT dataset is designed to evaluate models' ability to generalize to new visual compositions. It has two splits: Split A only contains gray, blue, brown and yellow cubes, but red, green, purple, and cyan cylinders; split B imposes the opposite color constraints on cubes and cylinders. If we directly learn visual concepts on split A, it overfits to classify shapes based on the color, leading to a poor generalization to split B.

Our solution is based on the idea of seeing attributes as operators. Specifically, we jointly train the concept embeddings (e.g., `Red`, `Cube`, etc.) as well as the semantic parser on split A, keeping pre-trained, frozen attribute operators. As we learn distinct representation spaces for different attributes, our model achieves an accuracy of 98.8% on split A and 98.9% on split B.

| Model | Prog. Anno. | Overall | Count | Cmp. Num. | Exist | Query Attr. | Cmp. Attr. |
|---|---|---|---|---|---|---|---|
| Human | N/A | 92.6 | 86.7 | 86.4 | 96.6 | 95.0 | 96.0 |
| NMN | 700K | 72.1 | 52.5 | 72.7 | 79.3 | 79.0 | 78.0 |
| N2NMN | 700K | 88.8 | 68.5 | 84.9 | 85.7 | 90.0 | 88.8 |
| IEP | 700K | 96.9 | 92.7 | 98.7 | 97.1 | 98.1 | 98.9 |
| DDRprog | 700K | 98.3 | 96.5 | 98.4 | 98.8 | 99.1 | 99.0 |
| TbD | 700K | **99.1** | **97.6** | **99.4** | **99.2** | **99.5** | **99.6** |
| RN | 0 | 95.5 | 90.1 | 93.6 | 97.8 | 97.1 | 97.9 |
| FiLM | 0 | 97.6 | 94.5 | 93.8 | 99.2 | 99.2 | 99.0 |
| MAC | 0 | 98.9 | 97.2 | **99.4** | **99.5** | 99.3 | **99.5** |
| NS-CL | 0 | **98.9** | **98.2** | 99.0 | 98.8 | **99.3** | 99.1 |

Table 4: Our model outperforms all baselines using no program annotations. It achieves comparable results with models trained by full program annotations such as TbD.

| Model | Test | | | |
|---|---|---|---|---|
| | Split A | Split B | Split C | Split D |
| MAC | 97.3 | N/A | 92.9 | N/A |
| IEP | 96.1 | 92.1 | 91.5 | 90.9 |
| TbD | 98.8 | 94.5 | 94.3 | 91.9 |
| NS-CL | **98.9** | **98.9** | **98.7** | **98.8** |

Figure 5: We test the combinatorial generalization w.r.t. the number of objects in scenes and the complexity of questions (i.e. the depth of the program trees). We makes four split of the data containing various complexities of scenes and questions. Our object-based visual representation and explicit program semantics enjoys the best (and almost-perfect) combinatorial generalization compared with strong baselines.

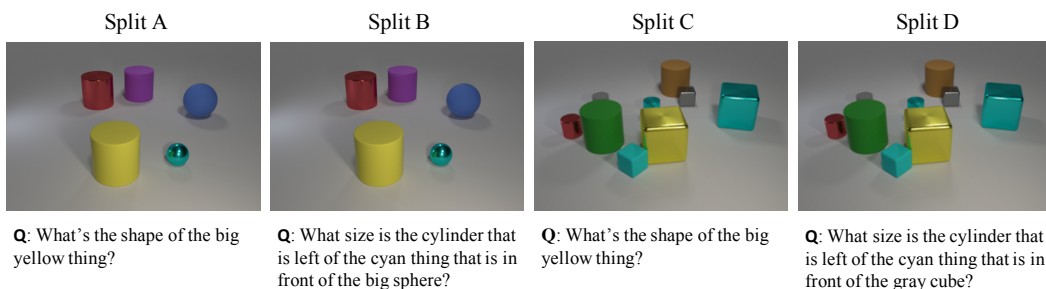

| Split A | Split B | Split C | Split D |
|---|---|---|---|
| **Q**: What's the shape of the big yellow thing? | **Q**: What size is the cylinder that is left of the cyan thing that is in front of the big sphere? | **Q**: What's the shape of the big yellow thing? | **Q**: What size is the cylinder that is left of the cyan thing that is in front of the gray cube? |

Figure 6: Samples collected from four splits in Section 4.3 for illustration. Models are trained on split A but evaluated on all splits for testing the combinatorial generalization.

**Generalizing to new visual concepts.** We expect the process of concept learning can take place in an incremental manner: having learned 7 different colors, humans can learn the 8-th color incrementally and efficiently. To this end, we build a synthetic split of the CLEVR dataset to replicate the setting of incremental concept learning. Split A contains only images without any purple objects, while split B contains images with at least one purple object. We train all the models on split A first, and finetune them on 100 images from split B. We report the final QA performance on split B's validation set. All models use a pre-trained semantic parser on the full CLEVR dataset.

Our model performs a 93.9% accuracy on the QA test in Split B, outperforming the convolutional baseline IEP (Johnson et al., 2017b) and the attentional baseline TbD (Mascharka et al., 2018) by 4.6% and 6.1% respectively. The acquisition of `Color` operator brings more efficient learning of new visual concepts.

## 4.4 Combinatorial generalization to new scenes and questions

Having learned visual concepts on small-scale scenes (containing only few objects) and simple questions (only single-hop questions), we humans can easily generalize the knowledge to larger-scale scenes and to answer complex questions. To evaluate this, we split the CLEVR dataset into four parts: **Split A** contains only scenes with less than 6 objects, and questions whose latent programs having a depth less than 5; **Split B** contains scenes with less than 6 objects, but arbitrary questions; **Split C** contains arbitrary scenes, but restricts the program depth being less than 5; **Split D** contains arbitrary scenes and questions. Figure 6 shows some illustrative samples.

As VQA baselines are unable to count a set of objects of arbitrary size, for a fair comparison, all programs containing the "count" operation over > 6 objects are removed from the set. For

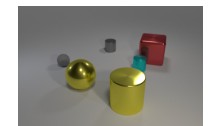

**Caption**: There is a big yellow cylinder in front of a gray object.

(a) An illustrative pair of image and caption in our synthetic dataset.

| Model | Retrieval Accuracy |
|---|---|
| IEP | 95.5 |
| TbD | **97.0** |
| NS-CL | 96.9 |

(b) Image-caption retrieval accuracy on a subset of data. Our model archives comparable results with VQA baselines.

| Model | Retrieval Accuracy |
|---|---|
| CNN-LSTM | 68.9 |
| NS-CL | **97.0** |

(c) Image-caption retrieval accuracy on the full dataset. Our model outperforms baselines and requires no extra training or fine-tuning of the visual perception module.

Table 5: We introduce a new simple DSL for image-caption retrieval to evaluate how well the learned visual concepts transfer. Due to the difference between VQA and caption retrieval, VQA baselines are only able to infer the result on a partial set of data. The learned object-based visual concepts can be directly transferred into the new domain for free.

methods using explicit program semantics, the semantic parser is pre-trained on the full dataset and fixed. Methods with implicit program semantics (Hudson & Manning, 2018) learn an entangled representation for perception and reasoning, and cannot trivially generalize to more complex programs. We only use the training data from the Split A and then quantify the generalization ability on other three splits. Shown in Table 5, our NS-CL leads to almost-perfect generalization to larger scenes and more complex questions, outperforming all baselines by at least 4% in QA accuracy.

## 4.5 EXTENDING TO OTHER PROGRAM DOMAIN

The learned visual concepts can also be used in other domains such as image retrieval. With the visual scenes fixed, the learned visual concepts can be directly transferred into the new domain. We only need to learn the semantic parsing of natural language into the new DSL.

We build a synthetic dataset for image retrieval and adopt a DSL from scene graph–based image retrieval (Johnson et al., 2015). The dataset contains only simple captions: "There is an <object A> <relation> <object B>." (e.g., There is a box right of a cylinder). The semantic parser learns to extract corresponding visual concepts (e.g., `box`, `right`, and `cylinder`) from the sentence. The program can then be executed on the visual representation to determine if the visual scene contains such relational triples.

For simplicity, we treat retrieval as classifying whether a relational triple exists in the image. This functionality cannot be directly implemented on the CLEVR VQA program domain, because questions such as "Is there a box right of a cylinder" can be ambiguous if there exist multiple cylinders in the scene. Due to the entanglement of the visual representation with the specific DSL, baselines trained on CLEVR QA can not be directly applied to this task. For a fair comparison with them, we show the result in Table 5b on a subset of the generated image-caption pairs where the underlying programs have no ambiguity regarding the reference of object B. A separate semantic parser is trained for the VQA baselines, which translates captions into a CLEVR QA-compatible program (e.g., `Exist(Filter(Box, Relate(Right, Filter(Cylinder))))`.

Table 5c compares our NS-CL against typical image-text retrieval baselines on the full image-caption dataset. Without any annotations of the sentence semantics, our model learns to parse the captions into the programs in the new DSL. It outperforms the CNN-LSTM baseline by 30%.

## 4.6 EXTENDING TO NATURAL IMAGES AND LANGUAGE

We further conduct experiments on MS-COCO (Lin et al., 2014) images. Results are presented on the VQS dataset (Gan et al., 2017). VQS contains a subset of images and questions from the original VQA 1.0 dataset (Antol et al., 2015). All questions in the VQS dataset can be visually grounded: each question is associated with multiple image regions, annotated by humans as essential for answering the question. Figure 7 illustrates an execution trace of NS-CL on VQS.

We use a syntactic dependency parser to extract programs and concepts from language (Andreas et al., 2016; Schuster et al., 2015). The object proposals and features are extracted from models pre-trained on the MS-COCO dataset and the ImageNet dataset, respectively. Illustrated in Figure 7, our model

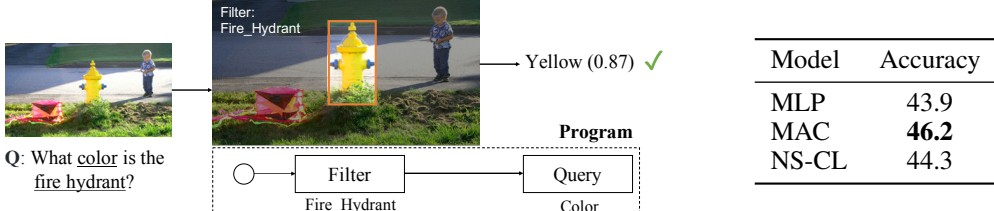

| Model | Accuracy |
|-------|----------|
| MLP   | 43.9     |
| MAC   | **46.2** |
| NS-CL | 44.3     |

Figure 7: **Left**: An example image-question pair from the VQS dataset and the corresponding execution trace of NS-CL. **Right**: Results on the VQS test set. Our model achieves a comparable results with the baselines.

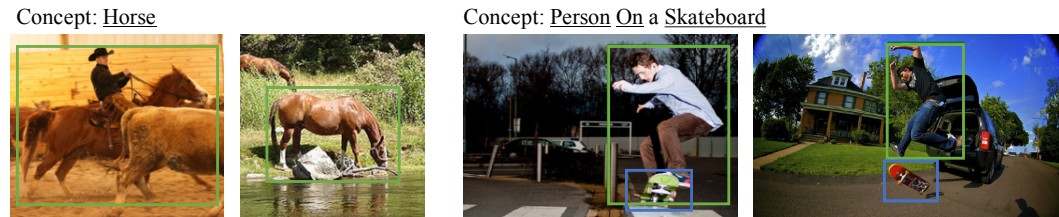

Figure 8: Concepts learned from VQS, including object categories, attributes, and relations.

shows competitive performance on QA accuracy, comparable with the MLP baseline (Jabri et al., 2016) and the MAC network (Hudson & Manning, 2018). Additional illustrative execution traces of NS-CL are in Appendix H. Beyond answering questions, NS-CL effectively learns visual concepts from data. Figure 8 shows examples of the learned visual concepts, including object categories, attributes, and relations. Experiment setup and implementation details are in Appendix G.2.

In this paper, we focus on a neuro-symbolic framework that learns visual concepts about object properties and relations. Indeed, visual question answering requires AI systems to reason about more general concepts such as events or activities (Levin, 1993). We leave the extension of NS-CL along this direction and its application to general VQA datasets (Antol et al., 2015) as future work.

## 5 DISCUSSION AND FUTURE WORK

We presented a method that jointly learns visual concepts, words, and semantic parsing of sentences from natural supervision. The proposed framework, NS-CL, learns by looking at images and reading paired questions and answers, without any explicit supervision such as class labels for objects. Our model learns visual concepts with remarkable accuracy. Based upon the learned concepts, our model achieves good results on question answering, and more importantly, generalizes well to new visual compositions, new visual concepts, and new domain specific languages.

The design of NS-CL suggests multiple research directions. First, constructing 3D object-based representations for realistic scenes needs further exploration (Anderson et al., 2018; Baradel et al., 2018). Second, our model assumes a domain-specific language for describing formal semantics. The integration of formal semantics into the processing of complex natural language would be meaningful future work (Artzi & Zettlemoyer, 2013; Oh et al., 2017). We hope our paper could motivate future research in visual concept learning, language learning, and compositionality.

Our framework can also be extended to other domains such as video understanding and robotic manipulation. Here, we would need to discover semantic representations for actions and interactions (e.g., push) beyond static spatial relations. Along this direction, researchers have studied building symbolic representations for skills (Konidaris et al., 2018) and learning instruction semantics from interaction (Oh et al., 2017) in constrained setups. Applying neuro-symbolic learning frameworks for concepts and skills would be meaningful future work toward robotic learning in complex interactive environments.

**Acknowledgements.** We thank Kexin Yi, Haoyue Shi, and Jon Gauthier for helpful discussions and suggestions. This work was supported in part by the Center for Brains, Minds and Machines (NSF STC award CCF-1231216), ONR MURI N00014-16-1-2007, MIT-IBM Watson AI Lab, and Facebook.

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

## A  CLEVR DOMAIN-SPECIFIC LANGUAGE AND IMPLEMENTATIONS

We first introduce the domain-specific language (DSL) designed for the CLEVR VQA dataset (Johnson et al., 2017a). Table 6 shows the available operations in the DSL, while Table 7 explains the type system.

| Operation | Signature | Semantics |
|---|---|---|
| Scene | () ⟶ ObjectSet | Return all objects in the scene. |
| Filter | (ObjectSet, ObjConcept) ⟶ ObjectSet | Filter out a set of objects having the object-level concept (e.g., red) from the input object set. |
| Relate | (Object, RelConcept) ⟶ ObjectSet | Filter out a set of objects that have the relational concept (e.g., left) with the input object. |
| AERelate | (Object, Attribute) ⟶ ObjectSet | (Attribute-Equality Relate) Filter out a set of objects that have the same attribute value (e.g., same color) as the input object. |
| Intersection | (ObjectSet, ObjectSet) ⟶ ObjectSet | Return the intersection of two object sets. |
| Union | (ObjectSet, ObjectSet) ⟶ ObjectSet | Return the union of two object sets. |
| Query | (Object, Attribute) ⟶ ObjConcept | Query the attribute (e.g., color) of the input object. |
| AEQuery | (Object, Object, Attribute) ⟶ Bool | (Attribute-Equality Query) Query if two input objects have the same attribute value (e.g., same color). |
| Exist | (ObjectSet) ⟶ Bool | Query if the set is empty. |
| Count | (ObjectSet) ⟶ Integer | Query the number of objects in the input set. |
| CLessThan | (ObjectSet, ObjectSet) ⟶ Bool | (Counting LessThan) Query if the number of objects in the first input set is less than the one of the second set. |
| CGreaterThan | (ObjectSet, ObjectSet) ⟶ Bool | (Counting GreaterThan) Query if the number of objects in the first input set is greater than the one of the second set. |
| CEqual | (ObjectSet, ObjectSet) ⟶ Bool | (Counting Equal) Query if the number of objects in the first input set is the same as the one of the second set. |

Table 6: All operations in the domain-specific language for CLEVR VQA.

We note that some function takes Object as its input instead of ObjectSet. These functions require the uniqueness of the referral object. For example, to answer the question "What's the color of the red object?", there should be one and only one red object in the scene. During the program execution, the input object set will be implicitly cast to the single object (if the set is non-empty and there is only one object in the set). Such casting is named Unique in related works (Johnson et al., 2017b).

| Type | Example | Semantics |
|------|---------|-----------|
| ObjConcept | `Red`, `Cube`, etc. | Object-level concepts. |
| Attribute | `Color`, `Shape`, etc. | Object-level attributes. |
| RelConcept | `Left`, `Front`, etc. | Relational concepts. |
| Object | ● | A single object in the scene. |
| ObjectSet | {●,■} | A set of objects in the scene. |
| Integer | $0, 1, 2, \cdots$ | A single integer. |
| Bool | `True`, `False` | A single boolean value. |

Table 7: The type system of the domain-specific language for CLEVR VQA.

## B  SEMANTIC PARSING

As shown in Appendix A, a program can be viewed as a hierarchy of operations which take concepts as their parameters. Thus, NS-CL generates the hierarchies of latent programs in a sequence to tree manner (Dong & Lapata, 2016). The semantic parser adopts an encoder-decoder architecture, which contains four neural modules: (1) a bidirectional GRU encoder `IEncoder` (Cho et al., 2014) to encode an input question into a fixed-length embedding, (2) an operation decoder `OpDecoder` that determines the operation tokens, such as `Filter`, in the program based on the sentence embedding, (3) a concept decoder `ConceptDecoder` that selects concepts appeared in the input question as the parameters for certain operations (e.g., `Filter` takes an object-level concept parameter while `Query` takes an attribute), and (4) a set of output encoders $\{$`OEncoder`$_i\}$ which encode the decoded operations by `OpDecoder` and output the latent embedding for decoding the next operation. The operation decoder, the concept decoder, and the output encoders work jointly and recursively to generate the hierarchical program layout. Algorithm 1 illustrates the algorithmic outline of the semantic parser.

---

**Algorithm 1:** The String-to-Tree Semantic Parser.

**Function** `parse`$(f, \{c_i\})$**:**
    $program \leftarrow$ `EmptyProgram()`;
    $program.op \leftarrow$ `OpDecoder`$(f)$;
    **if** $program.op$ requires a concept parameter **then**
        $program.concept \leftarrow$ `ConceptDecoder`$(f, \{c_i\})$;
    **for** $i = 0, 1, \cdots$ number of non-concept inputs of $program.op$ **do**
        $program.input[i] \leftarrow$ `parse` ( `OEncoder`$_i(f, program.op)$ ,$\{c_i\}$ );
    **return** $program$

---

The function $parse$ takes two inputs: the current decoding state $f$ and all concepts appeared in the question, as a set $\{c_i\}$. The parsing procedure begins with encoding the input question by `IEncoder` as $f_0$, extracting the concept set $\{c_i\}$ from the input question, and invoking $parse(f_0, \{c_i\})$.

The concept set $\{c_i\}$ is extracted using hand-coded rules. We assume that each concept (including object-level concepts, relational concepts, and attributes) is associated with a single word in the question. For example, the word "red" is associated with the object-level concept `Red`, while the word "shape" is associated with the attribute `Shape`. Informally, we call these words *concept words*. For a given question $Q$, the corresponding concept set $\{c_i\}$ is composed of all occurrences of the *concept words* in $Q$. The set of *concept words* is known for the CLEVR dataset. For natural language questions, one could run POS tagging to find all *concept words* (Andreas et al., 2016; Schuster et al., 2015). We leave the automatic discovery of concept words as a future work (Gauthier et al., 2018). We use the word embedding of the *concept words* as the representation for the concepts $\{c_i\}$. Note that, these "concept embeddings" are only for the program parsing. The visual module has separate concept embeddings for aligning object features with concepts in the visual-semantic space.

We now delve into the main function $parse(f, \{c_i\})$: we first decode the root operation $op$ of the hierarchy by $\texttt{OpDecoder}(f)$. If $op$ requires a concept parameter (an object-level concept, a relational concept, or an attribute), $\texttt{ConceptDecoder}$ will be invoked to choose a concept from all concepts $\{c_i\}$. Assuming $op$ takes two non-concept inputs (e.g., the operation $\texttt{Intersection}$ takes two object sets as its input), there will be two branches for this root node. Thus, two output encoders $\texttt{OEncoder}_0$ and $\texttt{OEncoder}_1$ will be applied to transform the current state $f$ into two sub-states $f_1$ and $f_2$. $parse$ will be recursively invoked based on $f_1$ and $f_2$ to generate the two branches respectively. In the DSL, the number of non-concept inputs for any operation is at most 2.

In our implementation, the input encoder $\texttt{IEncoder}$ first maps each word in the question into an embedding space. The word embeddings are composed of two parts: a randomly initialized word embedding of dimension 256 and a positional embedding of dimension 128 (Gehring et al., 2017). For a *concept word*, its word embedding only depends on which type it belongs to (i.e. object-level, relational or attribute). Thus, after being trained on a fixed dataset, the semantic parser can parse questions with novel (unseen) *concept words*. The sequence of word embeddings is then encoded by a two-layer GRU with a hidden dimension of $256 * 2$ (bidirectional). The function $parse$ starts from the last hidden state of the GRU, and works recursively to generate the hierarchical program layout. Both $\texttt{OpDecoder}$ and $\texttt{ConceptDecoder}$ are feed-forward networks. $\texttt{ConceptDecoder}$ performs attentions over the representations of all concepts $\{c_i\}$ to select the concepts. Output encoders $\texttt{OEncoder}_0$ and $\texttt{OEncoder}_1$ are implemented as GRU cells.

Another pre-processing of the sentence is to group consecutive object-level *concept words* into a group and treat them together as a single concept, inspired by the notion of "noun phrases" in natural languages. The computational intuition behind this grouping is that, the latent programs of CLEVR questions usually contain multiple consecutive $\texttt{Filter}$ tokens. During the program parsing and execution, we aim to fuse all such $\texttt{Filters}$ into a single $\texttt{Filter}$ operation that takes multiple concepts as its parameter.

**A Running Example**   As a running example, consider again the question "What is the color of the cube right of the red matte object?". We first process the sentence (by rules) as: "What is the <Attribute 1 (color)> of the <(ObjConcept 1 (cube)> <RelConcept 1 (right)> of the <ObjConcept 2 (red matte object)>?". The expected parsing result of this sentence is:

```
Query(<Attribute 1>,
     Filter(<ObjConcept 1>,
          Relate(<RelConcept 1>,
               Filter(<ObjConcept 2>, Scene)
          )
     )
).
```

The semantic parser encode the word embeddings with $\texttt{IEncoder}$. The last hidden state of the GRU will be used as $f_0$. The word embeddings of the *concept words* form the set $\{c_i\} = \{$Attribute 1, ObjConcept 1, RelConcept 1, ObjConcept 2$\}$. The function $parse$ is then invoked recursively to generate the hierarchical program layout. Table 8 illustrates the decoding process step-by-step.

## C  PROGRAM EXECUTION

In this section, we present the implementation of all operations listed in Table 6. We start from the implementation of Object-typed and ObjectSet-typed variables. Next, we discuss how to classify objects by object-level concepts or relational concept, followed by the implementation details of all operations.

**Object-typed and ObjectSet-typed variables.**   We consider a scene with $n$ objects. An Object-typed variable can be represented as a vector $\mathrm{Object}$ of length $n$, where $\mathrm{Object}_i \in [0, 1]$ and $\sum_i \mathrm{Object}_i = 1$. $\mathrm{Object}_i$ can be interpreted as the probability that the $i$-th object of the scene is being referred to. Similarly, an ObjectSet-typed variable can be represented as a vector $\mathrm{ObjectSet}$ of length $n$, where $\mathrm{ObjectSet}_i \in [0, 1]$. $\mathrm{ObjectSet}_i$ can be interpreted as the probability that the $i$-the object is in the set. To cast an ObjectSet-typed variable $\mathrm{ObjectSet}$ as an Object-typed variable

| Step | Inputs | Outputs | Recursive Invocation |
|------|--------|---------|---------------------|
| 1 | $f_0$ | OpDecoder$(f_0) \rightarrow$ Query; 
 ConceptDecoder$(f_0) \rightarrow <$ Attribute 1 $>$; 
 OEncoder$_0(f_0,$ Query$) \rightarrow f_1$ | $parse(f_1)$ |
| 2 | $f_1$ | OpDecoder$(f_1) \rightarrow$ Filter; 
 ConceptDecoder$(f_1) \rightarrow <$ ObjConcept 1 $>$; 
 OEncoder$_0(f_1,$ Filter$) \rightarrow f_2$ | $parse(f_2)$ |
| 3 | $f_2$ | OpDecoder$(f_2) \rightarrow$ Relate; 
 ConceptDecoder$(f_2) \rightarrow <$ RelConcept 1 $>$; 
 OEncoder$_0(f_2,$ Relate$) \rightarrow f_3$ | $parse(f_3)$ |
| 4 | $f_3$ | OpDecoder$(f_3) \rightarrow$ Filter; 
 ConceptDecoder$(f_3) \rightarrow <$ ObjConcept 2 $>$; 
 OEncoder$_0(f_3,$ Filter$) \rightarrow f_4$ | $parse(f_4)$ |
| 5 | $f_4$ | OpDecoder$(f_3) \rightarrow$ Scene; | (End of branch.) |

Table 8: A step-by-step running example of the recursive parsing procedure. The parameter $\{c_i\}$ is omitted for better visualization.

Object (i.e., the Unique operation), we compute: Object $=$ softmax$(\sigma^{-1}($ObjectSet$))$, where $\sigma^{-1}(x) = \log(x/(1-x))$ is the logit function.

**Concept quantization.** Denote $o_i$ as the visual representation of the $i$-th object, $OC$ the set of all object-level concepts, and $A$ the set of all object-level attributes. Each object-level concept $oc$ (e.g., Red) is associated with a vector embedding $v^{oc}$ and a L1-normalized vector $b^{oc}$ of length $|A|$. $b^{oc}$ represents which attribute does this object-level concept belong to (e.g., the concept Red belongs to the attribute Color). All attributes $a \in A$ are implemented as neural operators, denoted as $u^a$ (e.g., $u^{\texttt{Color}}$). To classify the objects as being Red or not, we compute:

$$\Pr[\text{object } i \text{ is Red}] = \sigma \left( \sum_{a \in A} \left( b_a^{\text{Red}} \cdot \frac{\langle u^a(o_i), v_{\text{Red}} \rangle - \gamma}{\tau} \right) \right),$$

where $\sigma$ denotes the Sigmoid function, $\langle \cdot, \cdot \rangle$ the cosine distance between two vectors. $\gamma$ and $\tau$ are scalar constants for scaling and shifting the values of similarities. By applying this classifier on all objects we will obtain a vector of length $n$, denoted as ObjClassify(Red). Similarly, such classification can be done for relational concepts such as Left. This will result in an $n \times n$ matrix RelClassify(Left), where RelClassify(Left)$_{j,i}$ is the probability that the object $i$ is left of the object $j$.

To classify whether two objects have the same attribute (e.g., have the same Color), we compute:

$$\Pr[\text{object } i \text{ has the same Color as object } j] = \sigma \left( \frac{\langle u^{\texttt{Color}}(o_i), u^{\texttt{Color}}(o_j) \rangle - \gamma}{\tau} \right),$$

We can obtain a matrix AEClassify(Color) by applying this classifier on all pairs of objects, where AEClassifier(Color)$_{j,i}$ is the probability that the object $i$ and $j$ have the same Color.

**Quasi-symbolic program execution.** Finally, Table 9 summarizes the implementation of all operators. In practice, all probabilities are stored in the log space for better numeric stability.

# D OPTIMIZATION OF THE SEMANTIC PARSER

To tackle the optimization in a non-smooth program space, we apply an off-policy program search process (Sutton et al., 2000) to facilitate the learning of the semantic parser. Denote $\mathbb{P}(s)$ as the set of all valid programs in the CLEVR DSL for the input question $s$. We want to compute the gradient w.r.t. $\Theta_s$, the parameters of the semantic parser:

$$\nabla_{\Theta_s} = \nabla_{\Theta_s} \mathbb{E}_P [r \cdot \log \Pr[P]],$$

| Signature | Implementation |
|---|---|
| `Scene`$() \to out$: ObjectSet | $out_i := 1$ |
| `Filter`$(in$: ObjectSet, $oc$: ObjConcept) $\to$ $\quad out$: ObjectSet | $out_i := \min(in_i, \text{ObjClassify}(oc)_i)$ |
| `Relate`$(in$: Object, $rc$: RelConcept) $\to$ $\quad out$: ObjectSet | $out_i := \sum_j (in_j \cdot \text{RelClassify}(rc)_{j,i})$ |
| `AERelate`$(in$: Object, $a$: Attribute) $\to$ $\quad out$: ObjectSet | $out_i := \sum_j (in_j \cdot \text{AEClassify}(a)_{j,i})$ |
| `Intersection`$(in^{(1)}$: ObjectSet, $\quad in^{(2)}$: ObjectSet) $\to out$: ObjectSet | $out_i := \min(in_i^{(1)}, in_i^{(2)})$ |
| `Union`$(in^{(1)}$: ObjectSet, $in^{(2)}$: ObjectSet) $\to$ $\quad out$: ObjectSet | $out_i := \max(in_i^{(1)}, in_i^{(2)})$ |
| `Query`$(in$: Object, $a$: Attribute) $\to$ $\quad out$: ObjConcept | $\Pr[out = oc] := \sum_i in_i \cdot \dfrac{\text{ObjClassify}(oc)_i \cdot b_a^{oc}}{\sum_{oc'} \text{ObjClassify}(oc')_i \cdot b_a^{oc'}}$ |
| `AEQuery`$(in^{(1)}$: Object, $in^{(2)}$: Object, $\quad a$: Attribute) $\to b$: Bool | $b := \sum_i \sum_j (in_i^{(1)} \cdot in_j^{(2)} \cdot \text{AEClassify}(a)_{j,i})$ |
| `Exist`$(in$: ObjectSet) $\to b$: Bool | $b := \max_i in_i$ |
| `Count`$(in$: ObjectSet) $\to i$: Integer | $i := \sum_i in_i$ |
| `CLessThan`$(in^{(1)}$: ObjectSet, $\quad in^{(2)}$: ObjectSet) $\to b$: Bool | $b := \sigma\big((\sum_i in_i^{(2)} - \sum_i in_i^{(1)} - 1 + \gamma_c)/\tau_c\big)$ |
| `CGreaterThan`$(in^{(1)}$: ObjectSet, $\quad in^{(2)}$: ObjectSet) $\to b$: Bool | $b := \sigma\big((\sum_i in_i^{(1)} - \sum_i in_i^{(2)} - 1 + \gamma_c)/\tau_c\big)$ |
| `CEqual`$(in^{(1)}$: ObjectSet, $\quad in^{(2)}$: ObjectSet) $\to b$: Bool | $b := \sigma\big((-|\sum_i in_i^{(1)} - \sum_i in_i^{(2)}| + \gamma_c)/(\gamma_c \cdot \tau_c)\big)$ |

Table 9: All operations in the domain-specific language for CLEVR VQA. $\gamma_c = 0.5$ and $\tau_c = 0.25$ are constants for scaling and shift the probability. During inference, one can quantify all operations as Yi et al. (2018).

where $P \sim \text{SemanticParse}(s; \Theta_s)$. In REINFORCE, we approximate this gradient via Monte Carlo sampling.

An alternative solution is to exactly compute the gradient. Note that in the definition of the reward $r$, only the set of programs $\mathbb{Q}(s)$ leading to the correct answer will contribute to the gradient term. With the perception module fixed, the set $\mathbb{Q}$ can be efficiently determined by an off-policy exhaustive search of all possible programs $\mathbb{P}(s)$. In the third stage of the curriculum learning, we search for the set $\mathbb{Q}$ offline based on the quantified results of concept classification and compute the exact gradient $\nabla \Theta_s$. An intuitive explanation of the off-policy search is that, we enumerate all possible programs, execute them on the visual representation, and find the ones leading to the correct answer. We use $\mathbb{Q}(s)$ as the "groundtruth" program annotation for the question, to supervise the learning, instead of running the Monte Carlo sampling-based REINFORCE.

**Spurious program suppression.** However, directly using $\mathbb{Q}(s)$ as the supervision by computing $\ell = \sum_{p \in \mathbb{Q}(S)} -\log \Pr(p)$ can be problematic, due to the spuriousness or the ambiguity of the programs. This comes from two aspects:
1) intrinsic ambiguity: two programs are different but equivalent. For example

> P1: `AEQuery(Color,Filter(Cube),Filter(Sphere))` and
> P2: `Exist(Filter(Sphere,AERelate(Color,Filter(Cube))))`

are equivalent.
2) extrinsic spuriousness: one of the program is incorrect, but also leads to the correct answer in a

specific scene. For example,

P1: $\texttt{Filter}(\texttt{Red},\texttt{Relate}(\texttt{Left},\texttt{Filter}(\texttt{Sphere})))$ and
P2: $\texttt{Filter}(\texttt{Red},\texttt{Relate}(\texttt{Left},\texttt{Filter}(\texttt{Cube})))$

may refer to the same red object in a specific scene. Motivated by the REINFORCE process, to suppress such spurious programs, we use the loss function:

$$\ell = \sum_{p\in\mathbb{Q}} \text{stop\_gradient}(\Pr[p]) \cdot (-\log \Pr[p]).$$

The corresponding gradient $\nabla_{\Theta_s}$ is,

$$\nabla_{\Theta_s} = \sum_{p\in\mathbb{Q}} \Pr[p] \cdot \nabla_{\Theta_s}\left(r \cdot \log \Pr[P]\right) = \nabla_{\Theta_s}\left(\sum_{p\in\mathbb{Q}} r \cdot \Pr[p]\right).$$

The key observation is that, given a sufficiently large set of scenes, a program can be identified as spurious if there exists at least one scene where the program leads to a wrong answer. As the training goes, spurious programs will get less update due to the sampling importance term $\Pr[p]$ which weights the likelihood maximization term.

## E  CURRICULUM LEARNING SETUP

During the whole training process, we gradually add more visual concepts and more complex question examples into the model. Summarized in Figure 4(A), in general, the whole training process is split into 3 stages. First, we only use questions from lesson 1 to let the model learn object-level visual concepts. Second, we train the model to parse simple questions and to learn relational concepts. In this step, we freeze the neural operators and concept embeddings of object-level concepts. Third, the model gets trained on the full question set (lesson 3), learning to understand questions of different complexities and various format. For the first several iterations in this step, we freeze the parameters in the perception modules. In addition, during the training of all stages, we gradually increase the number of objects in the scene: from 3 to 10.

We select questions for each lesson in the curriculum learning by their depth of the latent program layout. For eaxmple, the program "$\texttt{Query}(\texttt{Shape},\texttt{Filter}(\texttt{Red},\texttt{Scene}))$" has the depth of 3, while the program "$\texttt{Query}(\texttt{Shape},\texttt{Filter}(\texttt{Cube},\texttt{Relate}(\texttt{Left},\texttt{Filter}(\texttt{Red},\texttt{Scene}))))$" has the depth of 5. Since we have fused consecutive $\texttt{Filter}$ operations into a single one, the maximum depth of all programs is 9 on the CLEVR dataset. We now present the detailed split of our curriculum learning lessons:

For lesson 1, we use only programs of depth 3. It contains three types of questions: querying an attribute of the object, querying the existence of a certain type of objects, count a certain type of objects, and querying if two objects have the same attribute (e.g., of the same color). These questions are almost about fundamental object-based visual concepts. For each image, we generate 5 questions of lesson 1.

For lesson 2, we use programs of depth less than 5, containing a number of questions regarding relations, such as querying the attribute of an object that is left of another object. We found that in the original CLEVR dataset, all $\texttt{Relate}$ operations are followed by a $\texttt{Filter}$ operation. This setup degenerates the performance of the learning of relational concepts such as $\texttt{Left}$. Thus, we add a new question template into the original template set: $\texttt{Count}(\texttt{Relate}(\,\cdot\,,\texttt{Filter}(\,\cdot\,,\texttt{Scene})))$ (e.g., "What's the number of objects that are left of the cube?"). For each image, we generate 5 questions of lesson 2.

For lesson 3, we use the full CLEVR question set.

Curriculum learning is crucial for the learning of our neuro-symbolic concept learner. We found that by removing the curriculum setup w.r.t. the number of object in the scenes, the visual perception module will get stuck at an accuracy that is similar to a random-guess model, even if we only use stage-1 questions. If we remove the curriculum setup w.r.t. the complexity of the programs, the joint training of the visual perception module and the semantic parser can not converge.

# F    ABLATION STUDY

We conduct ablation studies on the accuracy of semantic parsing, the impacts of the ImageNet pre-training of visual perception modules, the data efficiency of our model, and the usage of object-based representations.

## F.1    SEMANTIC PARSING ACCURACY.

We evaluate how well our model recovers the underlying programs of questions. Due to the intrinsic equivalence of different programs, we evaluate the accuracy of programs by executing them on the ground-truth annotations of objects. Invalid or ambiguous programs are also considered as incorrect. Our semantic parser archives $> 99.9\%$ QA accuracy on the validation split.

## F.2    IMPACTS OF THE IMAGENET PRE-TRAINING.

The only extra supervision of the visual perception module comes from the pre-training of the perception modules on ImageNet (Deng et al., 2009). To quantify the influence of this pre-training, we conduct ablation experiments where we randomly initialize the perception module following He et al. (2015). The classification accuracies of the learned concepts almost remain the same except for `Shape`. The classification accuracy of `Shape` drops from 98.7 to 97.5 on the validation set while the overall QA accuracy on the CLEVR dataset drops to 98.2 from 98.9. We speculate that large-scale image recognition dataset can provide prior knowledge of shape.

## F.3    DATA EFFICIENCY AND OBJECT-BASED REPRESENTATIONS

In this section, we study whether and how the number of training samples and feature representations affect the overall performance of various models on the CLEVR dataset. Specifically, we compare the proposed NS-CL against two strong baselines: TbD (Mascharka et al., 2018) and MAC (Hudson & Manning, 2018).

**Baselines.**    For comparison, we implement two variants of the baseline models: TbD-Object and MAC-Object. Inspired by Anderson et al. (2018), instead of using a 2D convolutional feature map, TbD-Object and MAC-Object take a stack of object features as inputs, whose shape is $k \times d_{obj}$. $k$ is the number of objects in the scene, and $d_{obj}$ is the feature dimension for a single object. In our experiments, we fix $k = 12$ as a constant value. If there are fewer than 12 objects in the scene, we add "null" objects whose features are all-zero vectors.

We extract object features in the same way as NS-CL. Features are extracted from a pre-trained ResNet-34 network before the last residual block for a feature map with high resolution. For each object, its feature is composed of two parts: region-based (by RoI Align) and image-based features. We concatenate them to represent each object. As discussed, the inclusion of the representation of the full scene is essential for the inference of relative attributes such as size or spatial position on the CLEVR domain.

TbD and MAC networks are originally designed to use image-level attention for reasoning. Thus, we implement two more baselines: TbD-Mask and MAC-Mask. Specifically, we replace the original attention module on images with a mask-guided attention. Denotes the union of all object masks as $M$. Before the model applies the attention on the input image, we multiply the original attention map computed by the model with this mask $M$. The multiplication silences the attention on pixels that are not part of any objects.

**Results.**    Table 3 summarizes the results. We found that TbD-Object and MAC-Object approach show inferior results compared with the original model. We attribute this to the design of the network architectures. Take the `Relate` operation (e.g., finds all objects left of a specific object $x$) as an example. TbD uses a stack of dilated convolutional layers to propagate the attention from object $x$ to others. In TbD-Object, we replace the stack of 2D convolutions by several 1D convolution layers, operating over the $k \times d_{obj}$ object features. This ignores the equivalence of objects (the order of objects should not affect the results). In contrast, MAC networks always use the attention mechanism

to extract information from the image representation. This operation is invariant to the order of objects, but is not suitable for handling quantities (e.g., counting objects).

As for TbD-Mask and MAC-Mask, although the mask-guided attention does not improve the overall performance, we have observed noticeably faster convergence during model training. TbD-Mask and MAC-Mask leverage the prior knowledge of object masks to facilitate the attention. Such prior has also been verified to be effective in the original TbD model: TbD employs an attention regularization during training, which encourages the model to attend to smaller regions.

In general, NS-CL is more data-efficient than MAC networks and TbD. Recall that NS-CL answers questions by executing symbolic programs on the learned visual concepts. Only visual concepts (such as `Red` and `Left`) and the interpretation of questions (how to translate questions into executable programs) need to be learned from data. In contrast, both TbD and MAC networks need to additionally learn to execute (implicit or explicit) programs such as counting.

For the experiments on the full CLEVR training set, we split 3,500 images (5% of the training data) as the hold-out validation set to tune the hyperparameters and select the best model. We then apply this model to the CLEVR validation split and report the testing performance. Our model reaches an accuracy of 99.2% using the CLEVR training set.

# G    EXTENDING TO OTHER SCENE AND LANGUAGE DOMAINS

## G.1    MINECRAFT DATASET

We also extend the experiments to a new reasoning testbed: Minecraft worlds (Yi et al., 2018). The Minecraft reasoning dataset differs from CLEVR in both visual appearance and question types. Figure 9 gives an example instance from the dataset.

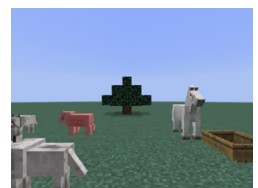

**Q**: What direction is the closest creature facing?
**A**: Left.
**P**: Query(Direction, FilterMost(Closest,
         Filter(Creature)
         ))

Figure 9: An example image and a related question-answering pair from the Minecraft dataset.

**Setup.**    Following Yi et al. (2018), we generate 10,000 Minecraft scenes using the officially open-sourced tools by Wu et al. (2017). Each image contains 3 to 6 objects. The objects are chosen from 12 categories, with 4 different facing directions (front, back, left and right). They stand on a 2D plane.

Besides different 3D visual appearance and image contexts, the Minecraft reasoning dataset introduces two new types of reasoning operations. We add them to our domain-specific language:

1. `FilterMost`(ObjectSet, Concept) → ObjectSet: Given a set of objects, finds the "most" one. For example, `FilterMost`(`Closest`, set) locates the object in the input set that is cloest to the camera (e.g., what is the direction of the closest animal?)
2. `BelongTo`(Object, ObjectSet) → Bool: Query if the input object belongs to a set.

**Results.**    Table 10 summarizes the results and Figure 12 shows sample execution traces. We compare our method against the NS-VQA baseline (Yi et al., 2018), which uses strong supervision for both scene representation (e.g., object categories and positions) and program traces. In contrast, our method learns both by looking at images and reading question-answering pairs. NS-CL outperforms NS-VQA by 5% in overall accuracy. We attribute the inferior results of NS-VQA to its derendering module. Because objects in the Minecraft world usually occlude with each other, the detected object bounding boxes are inevitably noisy. During the training of the derendering module, each detected bounding box is matched with one of the ground-truth bounding boxes and uses its class and pose as supervision. Poorly localized bounding boxes lead to noisy labels and hurt the accuracy of the derendering module. This further influences the overall performance of NS-VQA.

| Model | Overall | Count | Exist | Belong | Query |
|-------|---------|-------|-------|--------|-------|
| NS-VQA | 87.7 | 83.3 | 91.5 | 91.1 | 86.4 |
| NS-CL | **93.3** | **91.3** | **95.6** | **93.9** | **94.3** |

Table 10: Our model achieves comparable results on the Minecraft dataset with baselines trained by full program annotations.

### G.2  VQS DATASET

We conduct experiments on the VQS dataset (Gan et al., 2017). VQS is a subset of the VQA 1.0 dataset (Antol et al., 2015). It contains questions that can be visually grounded: each question is associated with multiple image regions, annotated by humans as necessary for answering the question.

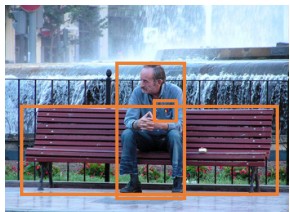

**Q**: Does this man have any pens on him?
**A**: Yes.
**P**: Exist(Filter(Man,
    Relate(Have, Filter(Pen))
    ))

Figure 10: An example image from the VQS dataset. The orange bounding boxes are object proposals. On the right, we show the original question and answer in natural language, as well as the latent program recovered by our parser. To answer this question, models are expected to attend to the man and his pen in the pocket.

**Setup.**  All models are trained on the first 63,509 images of the training set, and tested on the test split. For hyper-parameter tuning and model selection, the rest 5,000 images from the training set are used for validation. We use the multiple-choice setup for VQA: the models choose their most confident answer from 18 candidate answers for each question.

To obtain the latent programs from natural languages, we use a pre-trained syntactic dependency parser (Andreas et al., 2016; Schuster et al., 2015) for extracting programs and concepts that need to be learned. A sample question and the program obtained by our parser is shown in Figure 10. The concept embeddings are initialized by the bag of words (BoW) over the GloVe word embeddings (Pennington et al., 2014).

**Baselines.**  We compare our model against two representative baselines: MLP (Jabri et al., 2016) and MAC (Hudson & Manning, 2018).

MLP is a standard baseline for visual-question answering, which treats the multiple-choice task as a ranking problem. For a specific candidate answer, a multi-layer perceptron (MLP) model is used to encode a tuple of the image, the question, and the candidate answer. The MLP outputs a score for each tuple, and the answer to the question is the candidate with the highest score. We encode the image with a ResNet-34 pre-trained on ImageNet and use BoW over the GloVe word embeddings for the question and option encoding.

We slightly modify the MAC network for the VQS dataset. For each candidate answer, we concatenate the question and the answer as the input to the model. The MAC model outputs a score from 0 to 1 and the answer to the question is the candidate with the highest score. The image features are extracted from the same ResNet-34 model.

**Results.**  Table 7 summarizes the results. NS-CL achieves comparable results with the MLP baseline and the MAC network designed for visual reasoning. Our model also brings transparent reasoning over natural images and language. Example execution traces generated by NS-CL are shown in Figure 13. Besides, the symbolic reasoning process helps us to inspect the model and diagnose the error sources. See the caption for details.

## H  VISUALIZATION OF EXECUTION TRACES AND VISUAL CONCEPTS

Another appealing benefit is that our reasoning model enjoys full interpretability. Figure 11, Figure 12, and Figure 13 show NS-CL's execution traces on CLEVR, Minecraft, and VQS, respectively. As a side product, our system detects ambiguous and invalid programs and throws out exceptions. As an example (Figure 11), the question "What's the color of the cylinder?" can be ambiguous if there are multiple cylinders or even invalid if there are no cylinders.

Figure 14 and Figure 15 include qualitative visualizations of the concepts learned from the CLEVR and Minecraft datasets, including object categories, attributes, and relations. We choose samples from the validation or test split of each dataset by generating queries of the corresponding concepts. We set a threshold to filter the returned images and objects. For quantitative evaluations of the learned concepts on the CLEVR dataset, please refer to Table 2 and Table 5.

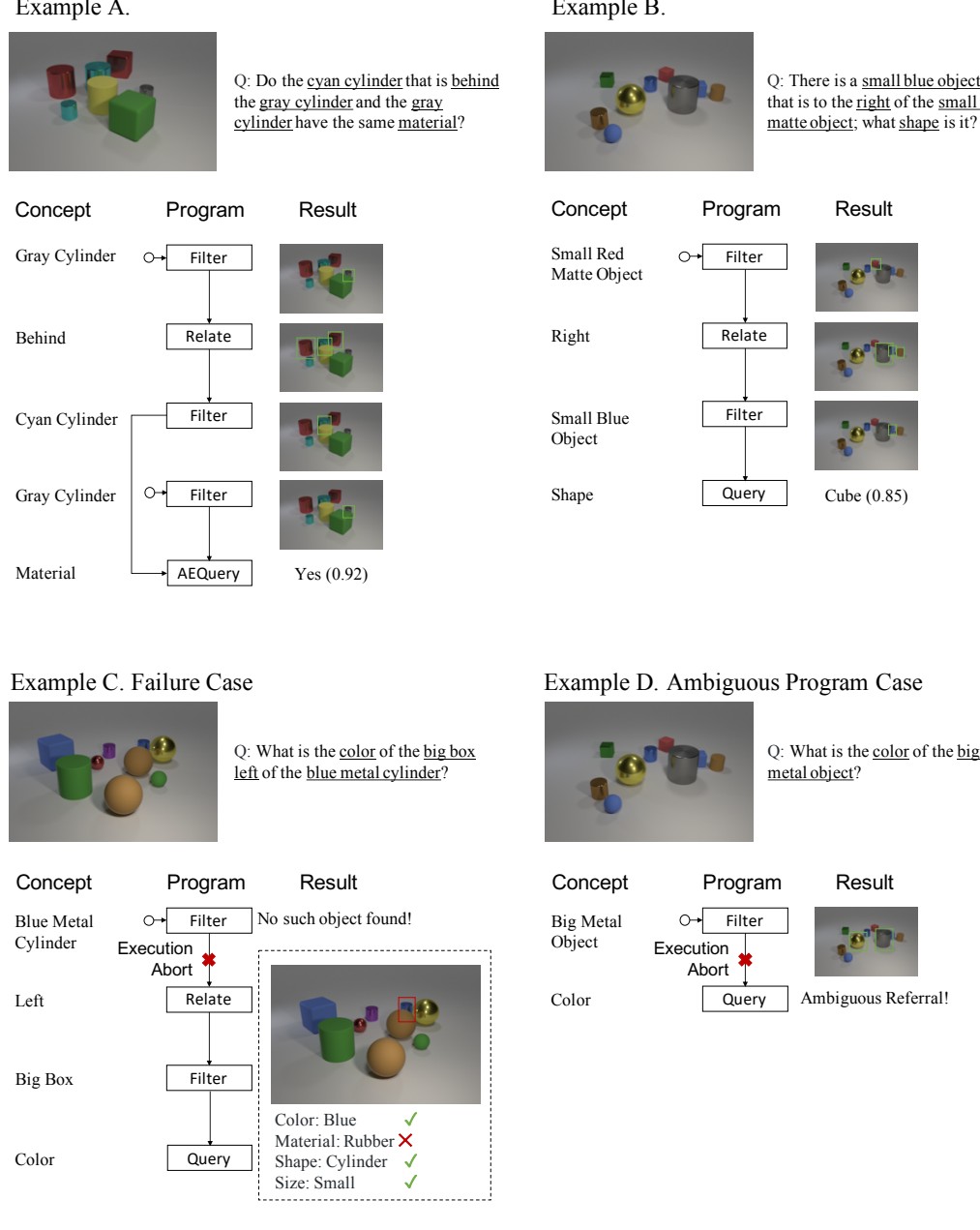

Figure 11: Visualization of the execution trace generated by our Neuro-Symbolic Concept Learner on the CLEVR dataset. Example A and B are successful executions that generate correct answers. In example C, the execution aborts at the first operator. To inspect the reason why the execution engine fails to find the corresponding object, we can read out the visual representation of the object, and locate the error source as the misclassification of the object material. Example D shows how our symbolic execution engine can detect invalid or ambiguous programs during the execution by performing sanity checks.

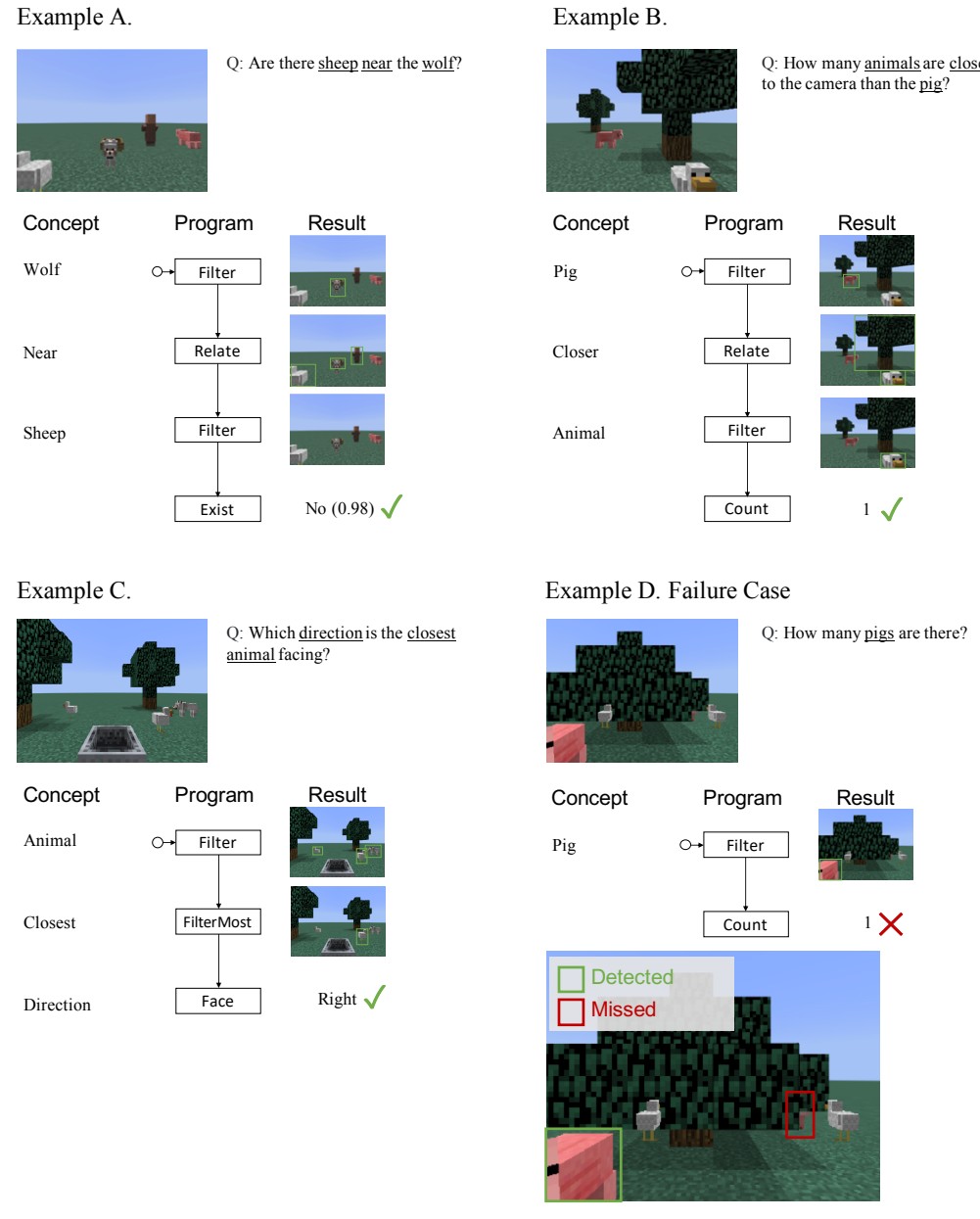

Figure 12: Exemplar execution trace generated by our Neuro-Symbolic Concept Learner on the Minecraft reasoning dataset. Example A, B and C are successful execution. Example C demonstrates the semantics of the FilterMost operation. Example D shows a failure case: the detection model fails to detect a pig hiding behind the big tree.

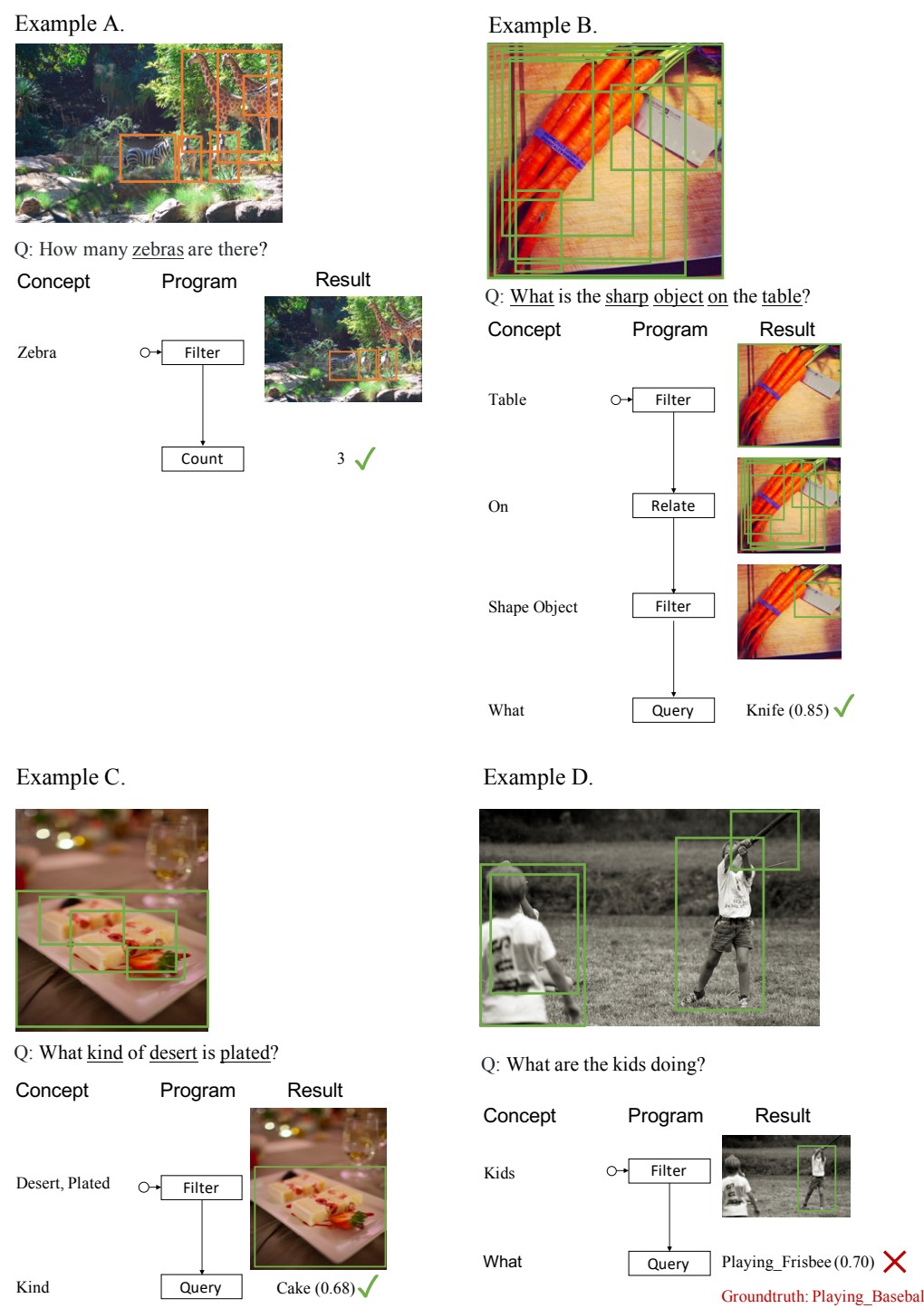

Figure 13: Illustrative execution trace generated by our Neuro-Symbolic Concept Learner on the VQS dataset. Execution traces A and B shown in the figure leads to the correct answer to the question. Our model effectively learns visual concepts from data. The symbolic reasoning process brings transparent execution trace and can easily handle quantities (e.g., object counting in Example A). In Example C, although NS-CL answers the question correctly, it locates the wrong object during reasoning: a dish instead of the cake. In Example D, our model misclassifies the sport as frisbee.

Concept: Cylinder

Concept: Matte

Concept: Blue Sphere

Concept: Yellow Object Left of Cylinder

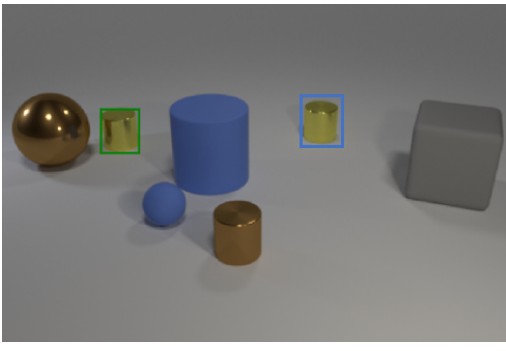 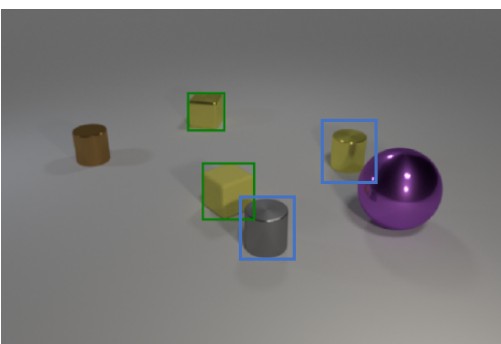

Figure 14: Concepts learned on the CLEVR dataset.

Concept: Wolf

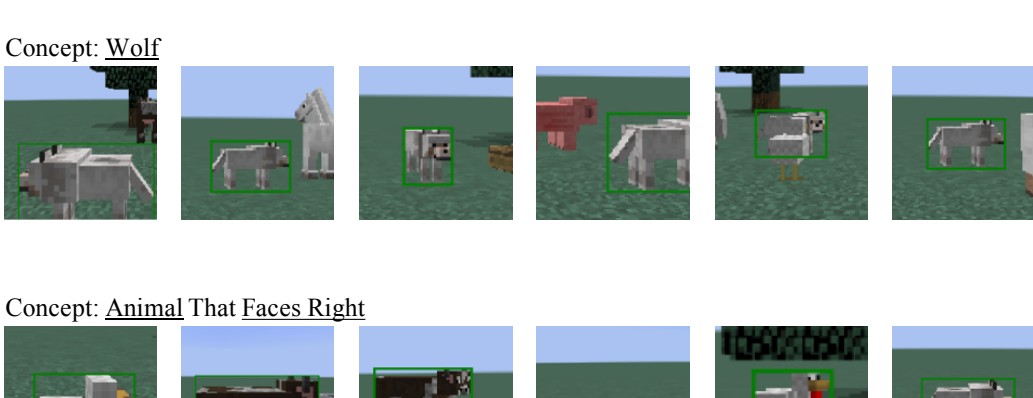

Concept: Animal That Faces Right

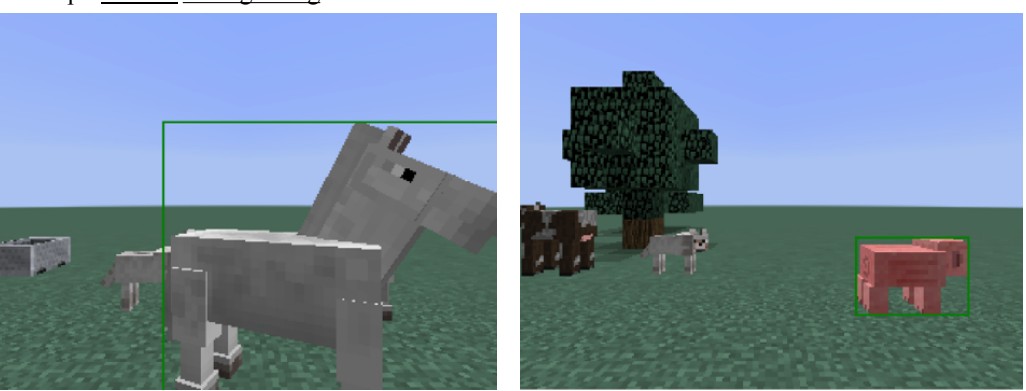

Concept: Closest Living Thing

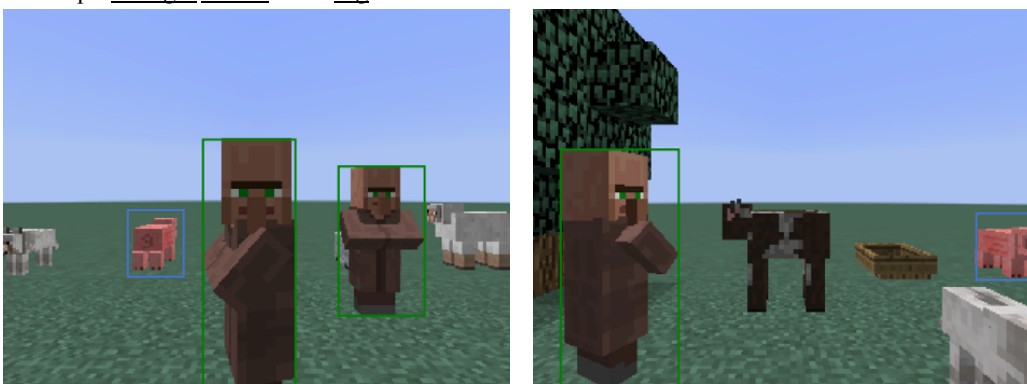

Concept: Villager Closer Than Pig

Figure 15: Concepts learned on the Minecraft dataset.

