# OpenReview forum: "The Neuro-Symbolic Concept Learner: Interpreting Scenes, Words, and Sentences From Natural Supervision"
_ICLR.cc/2019/Conference_

### Official Review · AnonReviewer1 · 2018-10-27
**Excellent paper including many cutting edge techniques**

**Rating:** 9
**Confidence:** 5

**Review:**

The paper is well written and flow well. The only thing I would like to see added is an elaboration of
"run a semantic parsing module to translate a question into an executable program". How to do semantic parsing is far from obvious. This topic needs at least a paragraph of its own.

This is not a requirement but an opportunity, can you explain how counting work? I think you have it at the standard level of the magic of DNN but some digging into the mechanism would be appreciated.

In concluding maybe you can speculate how far this method can go. Compositionality? Implicit relations inferred from words and behavior? Application to video with words?

---

> ### Author Response · Authors · 2018-11-15
> **Our Response to Reviewer 1**
>
> Thank you very much for the constructive comments.
>
> 1. Semantic parsing.
> In short, the semantic parsing module is a neural sequence-to-tree model. Given a natural language question, the module translates into an executable program with a hierarchy of primitive operation. We present an overview in Sec. 3.1 (last paragraph of Page 4), with more implementation details in Appendix B. We’ll revise the text for better clarity.
>
> The module begins with encoding the question into a fixed-length embedding vector using a bidirectional GRU. The decoder, taking the sentence embedding as input, recovers the hierarchy of the operations in a top-down manner: It first predicts the root token (the question type: query/count/… in the VQA case); then, conditioned on the root token, it predicts the tokens of the root’s children. The decoding algorithm runs recursively.
>
> 2. Counting.
> We perform counting in a quasi-symbolic manner, based on the object-based scene representation. As an example, consider a simple program: Count(Filter(Red)), which counts the number of red objects in the scene. The operation Filter(Red) assigns each object with a value p_i, as the confidence of classifying this object as a red one. Counting is performed as: $\sum_i p_i$. During inference, we round this value to the nearest integer. More details can be found in Sec. 3,1. (Page 5) and Appendix C. We will also revise the text for better clarity.
>
> Compared with alternatives, our method enjoys combinatorial generalization with the notion of `objects’: for example, trained on scenes with <= 6 objects, our model can also perform counting on scenes with 10 objects.
>
> 3. Future direction
> We thank the reviewer for the suggestions on future directions and will include the following discussions in the revision:
>
> Compositionality. We currently view the scene as a collection of objects with latent representations. Building scene (or video) representations that also reflects the compositional nature of objects (e.g., an object is a combination of multiple primitives) will be an interesting research direction.
>
> Infer relations from words and behavior. Modelling actions (e.g., push and pull) as concepts is another interesting direction. People have studied the symbolic representation of skills [1] and learning word (instruction) meanings from interaction [2].
>
> Videos and words. Our framework can also be extended to the video domain. Video techniques such as detection and tracking are needed to build the object-based representation [3]. Also, the semantic representation of sentences should be extended to include actions / interactions besides static spatial relations.
>
> We have also listed all other planned changes in our general response above. Please don’t hesitate to let us know for any additional comments on the paper or on the planned changes.
>
>
> [1] Konidaris, George, Leslie Pack Kaelbling, and Tomas Lozano-Perez. "From skills to symbols: Learning symbolic representations for abstract high-level planning." Journal of Artificial Intelligence Research 61 (2018): 215-289.
> [2] Oh, Junhyuk, Satinder Singh, Honglak Lee, and Pushmeet Kohli. "Zero-Shot Task Generalization with Multi-Task Deep Reinforcement Learning." In ICML, 2017.
> [3] Baradel, Fabien, Natalia Neverova, Christian Wolf, Julien Mille, and Greg Mori. "Object Level Visual Reasoning in Videos." In ECCV, 2018.

---

### Official Review · AnonReviewer3 · 2018-11-02
**Concern of invalid evaluation and vague demonstration of the contribution**

**Rating:** 6
**Confidence:** 4

**Review:**

To achieve the state-of-the-art on the CLEVR and the variations of this, the authors propose a method to use object-based visual representations and a differentiable quasi-symbolic executor. Since the semantic parser for a question input is not differentiable, they use REINFORCE algorithm and a technique to reduce its variance.

Quality:
The issue of invalid evaluation should be addressed. CLEVR dataset has train, validation, and test sets. Since the various hyper-parameters are determined with the validation set, the comparison of state-of-the-art should be done using test set. As the authors mentioned, REINFORCE algorithm may introduce high variance, this notion is critical to report valid results. However, the authors only report on the validation set in Table 2 including the main results, Table 4. For Table 5, they only specify train and test splits. Therefore, I firmly recommend the authors to report on the test set for the fair comparison with the other competitive models, and please describe how to determine the hyperparameters in all experimental settings.

Clarity:
As mentioned above, please specify the experimental details regarding setting hyperparameters.
In Experiments section, the authors used less than 10% of CLEVR training images. How about to use 100% of the training examples? How about to use the same amount of training examples in the competitive models? The report is incomplete to see the differential evident from the efficient usage of training examples.

Originality and significance:
The authors argue that object-based visual representation and symbolic reasoning are the contributions of this work (excluding the recent work, NS-VQA < 1 month). However, bottom-up and top-down attention work [1] shows that attention networks using object-based visual representation significantly improve VQA and image captioning performances. If the object-based visual representation alone is the primary source of improvement, it severely weakens the argument of the neuro-symbolic concept learner. Since, considering the trend of gains, the contribution of the proposing method seems to be incremental, this concern is inevitable. To defend this critic, the additional experiment to see the improvement of the other attentional model (e.g, TbD, MAC) using object-based visual representations, without any other annotations, is needed.

Pros:
- To confirm the effective learning of visual concepts, words, and semantic parsing of sentences, they insightfully exploit the nature of the CLEVR dataset for visual reasoning diagnosis.

Cons:
- Invalid evaluation to report only on the validation set, not test set.
- The unclear significance of the proposed method combining object-based visual representations and symbolic reasoning
- In the original CLEVR dataset paper, the authors said "we stress that accuracy on CLEVR is not an end goal in itself" and "..CLEVR should be used in conjunction with other VQA datasets in order to study the reasoning abilities of general VQA systems." Based on this suggestion, can this work generalize to real-world settings? This paper lacks to discuss its limitation and future direction toward the general problem settings.

Minor comments:
In 4.3, please fix the typos, "born" -> "brown" and "convlutional" -> "convolutional".


[1] Anderson, P., He, X., Buehler, C., Teney, D., Johnson, M., Gould, S., & Zhang, L. (2018). Bottom-Up and Top-Down Attention for Image Captioning and Visual Question Answering. IEEE Computer Vision and Pattern Recognition (CVPR'18).

---

> ### Author Response · Authors · 2018-11-15
> **Our Response to Reviewer 3 (Part 2)**
>
> 4. Specific Questions
> - Choice of hyperparameters.
> We use the open-sourced implementation of Mask-RCNN [5] to generate object proposals. For all the training processes described in the rest of the paper, we used learning rate 1e-3 with a weight decay of 5e-4. We decay the learning rate by a factor 0.1 after 60% of the designated training epochs. The REINFORCE optimizer uses a discount factor of 0.95. In the main text, the variance of REINFORCE means the variance of the gradient estimation but not the variance of the performance (accuracy). We will also add the standard deviation of the model performance in the revision.
>
> - Data-efficiency
> Thanks for the very nice suggestion. We have conducted a more systematic study on the data efficiency and will include it the revision. The results are
>
> Trained on 10% of the images:
> TbD: 54.2%.
> MAC: 67.3%.
> NS-CL: 98.9%.
>
> Trained on 100% of the images:
> TbD: 99.1%.
> MAC: 98.9%.
> NS-CL: 99.2%.
>
> These results demonstrate that our model is more data-efficient.
>
> We have also listed all other planned changes in our general response above. Please don’t hesitate to let us know for any additional comments on the paper or on the planned changes.
>
> [1] Anderson et al. "Bottom-up and top-down attention for image captioning and visual question answering." In CVPR, 2018.
> [2] Baradel et al. "Object Level Visual Reasoning in Videos." In ECCV, 2018.
> [3] Artzi, Yoav, and Zettlemoyer. "Weakly supervised learning of semantic parsers for mapping instructions to actions." TACL 1 (2013): 49-62.
> [4] Oh et al. "Zero-Shot Task Generalization with Multi-Task Deep Reinforcement Learning." In ICML, 2017.
> [5] https://github.com/roytseng-tw/Detectron.pytorch

---

> > ### Comment · AnonReviewer3 · 2018-11-20
> > **Still waiting for the results with fair comparisons w.r.t Author Feedback 2.**
> >
> > Sincerely thank you for the detailed explanations and comments for a constructive rebuttal.
> >
> > Re: 1. Train/test split
> > R1-1) I understand the current evaluation issue on CLEVR. Then, could you confirm that your hyperparameters are found using **ONLY** training split since you have used the validation split like the test split to compare state-of-the-art?
> > R1-2) Have you asked the authors of CLEVR regarding this issue? And, what's their response? I appreciate if you can cite the authors' reply in Appendix as a pointer to refer in future works.
> >
> > Re: 2. Object-based representations and baselines
> > R2) With the positive results, I would like to consider increasing my rating considering the authors' argument of fair comparison.

---

> > > ### Author Response · Authors · 2018-11-27
> > > **Thanks for your patience. Revision uploaded.**
> > >
> > > Dear reviewer, we have updated our paper with the promised results.
> > >
> > > 1. Train/Val/Test split:
> > > We have included the new results of NS-CL using 100% of the CLEVR training images. We use 95% of the training images for learning, and the remaining 5% for validation, hyper-parameter tuning, and model selection. Validation images are used only in testing. This further pushes the overall accuracy of NS-CL to 99.2% on the validation split. Please refer to Section 4.2 for the new results. We adopt the same strategy in all newly added experiments, including those on the Minecraft dataset and the VQS dataset. For the results using only 10% of the CLEVR training images, we simply used the training set accuracy for model selection.
> > >
> > > We have tried to contact the authors of the CLEVR dataset, and will be pleased to share further information regarding the test split upon receiving any responses.
> > >
> > > 2. Object-based Representations and Data Efficiency.
> > > Thank for suggesting the related work and the additional baselines. We have added additional experiments that incorporate object-based representations into TbD/MAC (Section 4.2). NS-CL achieves higher data efficiency. We believe that this comes from the full disentanglement of visual concept learning and symbolic reasoning: how to execute program instructions based on the learned concepts is programmed in NS-CL.
> > >
> > > Compared with the attention-based baselines, our use of symbolic programs enables better integration with object-based representations, e.g., in modelling relations and quantities. For the detailed implementation of the baselines, please refer to Appendix E.3.
> > >
> > > Thanks again for your comments.

---

> > > > ### Comment · AnonReviewer3 · 2018-11-30
> > > > **No reason not to raise the score**
> > > >
> > > > The authors sufficiently clarified the experimental procedures for fair comparisons what I had concerned. Although the work seems to be limited in natural images and language (VQS), I appreciate the authors to include in the paper for the future works.
> > > >
> > > > I decide to increase my rating by 1.

---

> ### Author Response · Authors · 2018-11-15
> **Our Response to Reviewer 3 (Part 1)**
>
> Thank you very much for the constructive comments.
>
> 1. Train/test split
> Our evaluation is valid and fair, because all previous papers have also reported results only on the validation set, and we follow the tradition in this paper. They did this because there are no ground-truth labels or evaluation servers provided for the CLEVR test split. Evaluation on the test split is therefore impossible. We agree that it’s important to ensure all evaluation valid, and we’ll include this clarification into the revision.
>
> 2. Object-based representations and baselines
> Thanks for the suggestion. We’ll cite and discuss the paper that used object-based visual representation. We will also add additional experiments that incorporate object-based representations into TbD/MAC: Instead of the image feature extracted from a ResNet, we change the input visual feature to the reasoning neural architecture to be an object-based representation as in [1]. Please let us know if you have any suggestion regarding the comparison.
>
> We also want to clarify that the object-based representation alone is not the main contribution of the paper. Instead, our key contribution is the integration of object-based representations and symbolic reasoning. Such combination helps us disentangle visual concept learning and language understanding, and has three advantages over alternatives, as explored in the paper:
>
> 1) Executing symbolic programs on object-based representations naturally facilitates complex reasoning that includes quantities (counting), comparisons, and relations. It also brings combinatorial generalization by design (Sec. 4.4): for example, trained on scenes with <= 6 objects, our model (but not the baselines) can also perform counting on scenes with 10 objects.
>
> 2) It fully disentangles the visual concept learning and reasoning: once the visual concepts are learned, they can be systematically evaluated (Sec. 4.1) and deployed in any visual-semantic applications (such as image caption retrieval, as shown in Sec. 4.5). In contrast, earlier methods like IEP, TbD, and MAC learn visual concepts and reasoning in an entangled manner and cannot be easily adapted to new problem domains (e.g., show in Table 6, VQA baselines are only able to infer the result on a partial set of the image-caption data).
>
> 3) Symbolic execution over the object space brings full transparency. One can easily trace back the error answer and even detect adversarial (ambiguous or wrong) questions (please refer to Appendix. E for some examples).
>
> 3. Limitation and future work
> We’d like to clarify that we are not targeting at a specific application such as VQA; instead, we want to build a system that learns accurate (Sec. 4.1), interpretable (Sec. 4.2), and transferrable (Sec. 4.5) concepts from natural supervision: images and question-answer pairs. To achieve this, we propose a novel framework that 1) disentangles the learning of both, but 2) bridges them with a reasoning module and 3) lets them bootstrap the learning of each other.
>
> Toward concept learning from realistic images and complex language, the current model design suggest multiple research directions. First, our model relies on object-based representations; constructing 3D object-based representations for realistic scenes (or videos) needs further exploration [1,2]. Second, our model assumes a domain-specific language for a formal description of semantics. The integration of formal semantics into the processing of complex natural language would be meaningful future work [3,4]. We hope our paper could motivate future research in visual concept learning, language learning, and compositionality.

---

### Official Review · AnonReviewer2 · 2018-11-03
**Interesting end-to-end joint learning of visual concepts and semantic parsing but  experiments are limiting**

**Rating:** 7
**Confidence:** 4

**Review:**


Summary:
=========
The paper proposes a joint learning of visual representation and word and semantic parsing of the sentences given paired images and paired Q/A with a model called neuro-symbolic concept learner using curriculum learning. The paper reads well and is easy to follow. The idea of jointly learning visual concepts and language is an important task. Human reasoning involves learning and recall from multiple moralities. The authors use the CLEVR dataset for evaluation.

Strength:
========
- Jointly learning the language parsing and visual representations indirectly from paired Q/A and paired images is interesting. Combining the visual learning with the visual questions answers by decomposing them into primitive symbolic operations and reasoning in symbolic space seems interesting.

- End-to-end learning of the visual concepts, Q/A decomposition into primitives and program execution was shown to be competitive to baseline methods.

Weakness:
=========
- Although, the joint learning and composition is interesting, the visual task is simplistic and it is not obvious how this would generalize into other complex VQA tasks.

- Experiments are not as rigorous as the discussion of the methods suggests. Evaluation on more datasets would have made the comparisons and drawn conclusions more stronger. Although CLEVR is suited for learning relational concepts from referential expressions, it is a toy dataset. Applicability of the proposed method on other realistic datasets would have made the paper more stronger.

---

> ### Author Response · Authors · 2018-11-15
> **Our Response to Reviewer 2**
>
> Thank you very much for the encouraging and constructive comments. We agree that generalizing to more complex visual domains would be essential for our task. In the revision, we will include the results of NS-CL on new datasets, including the VQA dataset of real-world images [1] and the Minecraft dataset used by Yi et al. [2].
>
> We have also listed all other planned changes in our general response above. Please don’t hesitate to let us know for any additional comments on the paper or on the planned changes.
>
> [1] Antol, Stanislaw, Aishwarya Agrawal, Jiasen Lu, Margaret Mitchell, Dhruv Batra, C. Lawrence Zitnick, and Devi Parikh. "Vqa: Visual question answering." In ICCV, 2015.
> [2] Yi, Kexin, Jiajun Wu, Chuang Gan, Antonio Torralba, Pushmeet Kohli, and Joshua B. Tenenbaum. "Neural-Symbolic VQA: Disentangling Reasoning from Vision and Language Understanding." In NIPS, 2018.

---

### Author Response · Authors · 2018-11-15
**Our General Response**

We thank all reviewers for their comments. In addition to the specific response below, here we summarize our goal and the changes planned to be included in the revision.

We study concept learning---discovering both visual concepts and language concepts from natural supervision (unannotated images and question-answer pairs). With these learned concepts, our model can solve many problems, such as image captioning, retrieval, as well as VQA. But here the ability to solve VQA is really a by-product, not our end goal---learning accurate (Sec. 4.1), interpretable (Sec. 4.2), and transferrable (Sec. 4.5) concepts.

We agree with the reviewers that it’s important to demonstrate how our model works on real images with more complex visual appearance. As suggested, we plan to include the following changes in the revision by Nov. 26 (the new official revision deadline, extended from Nov. 23):
- We will include quantitative and qualitative results on new datasets: the VQA dataset of real-world images [1] and the Minecraft dataset used by Yi et al. [2].
- We will add a systematic study regarding the data efficiency of our model, compared with other VQA baselines in Sec. 4.2.
- We will compare our model with other baselines (TbD and MAC) built upon the object-based representations.
- We will include additional discussions on limitation and future work.

Please don’t hesitate to let us know for any additional comments on the paper or on the planned changes.

[1] Antol, Stanislaw, Aishwarya Agrawal, Jiasen Lu, Margaret Mitchell, Dhruv Batra, C. Lawrence Zitnick, and Devi Parikh. "Vqa: Visual question answering." In ICCV, 2015.
[2] Yi, Kexin, Jiajun Wu, Chuang Gan, Antonio Torralba, Pushmeet Kohli, and Joshua B. Tenenbaum. "Neural-Symbolic VQA: Disentangling Reasoning from Vision and Language Understanding." In NIPS, 2018.

---

### Author Response · Authors · 2018-11-27
**General Response: Revision Uploaded**

We thank all reviewers for their constructive comments and have updated our paper accordingly. Please check out the new version!

Specific changes include

1) We have compared with additional baselines that incorporate object-based representation with attention-based methods (MAC/TbD). The results are in Section 4.2 and the implementation details are in Appendix E.3. The symbolic program execution module in NS-CL shows better utilization of object-based representations.

2) We provided a systematic analysis of data efficiency in Section 4.2. NS-CL achieves higher data efficiency by disentangling visual concept learning and program-based symbolic reasoning.

3) We added the results on a new visual reasoning testbed --- the Minecraft dataset. Results can be found in Appendix F.1.

4) We added both quantitative and qualitative results on the VQS dataset, composed of natural images from the COCO dataset and human-annotated question-answering pairs. Please kindly find these results in Section 4.6 and the implementation details in Appendix F.2. NS-CL achieves a comparable results with the baselines and learns visual concepts from the noisy inputs.

5) We have cited and discussed the suggested related work.

6) We have also included more discussions on future work.

Please don’t hesitate to let us know for any additional comments on the paper.

---

### Public Comment · ~Dzmitry_Bahdanau1 · 2019-04-15
**a few questions**

Hi, nice paper!

Two quick questions:
1) Can you elaborate how (if) the models learn how many attributes/concepts are there? E.g. in CLEVR there are 4 attributes that take 3, 4, 4, 8 values. Are these numbers learn by the model, or are they given? I read the appendix but I am still not sure I understand.
2) Do you by any chance plan to release the code?

---

### Meta-Review · Area_Chair1 · 2018-12-18
**Strong paper in an interesting new direction**

**Confidence:** 4
**Recommendation:** Accept (Oral)

**Metareview:**

Strong paper in an interesting new direction.
More work should be done in this area.